# Harnessing Feature Resonance under Arbitrary Target Alignment for Out-of-Distribution Node Detection

**Shenzhi Yang[1,2,5]**    **Junbo Zhao[1]**   **Sharon Li[3]**   **Shouqing Yang[1,2,5]**   **Dingyu Yang[1,2]**
**Xiaofang Zhang[4]**    **Haobo Wang[1,2,5]**

[1] Zhejiang University
[2] Hangzhou High-Tech Zone (Binjiang) Institute of Blockchain and Data Security
[3] Department of Computer Sciences, University of Wisconsin-Madison
[4] School of Computer Science and Technology, Soochow University
[5] Innovation and Management Center, School of Software Technology(Ningbo), Zhejiang University
Corresponding to: wanghaobo@zju.edu.cn

## Abstract

Out-of-distribution (OOD) node detection in graphs is a critical yet challenging task. Most existing approaches rely heavily on fine-grained labeled data to obtain a pre-trained supervised classifier, inherently assuming the existence of a well-defined pretext classification task. However, when such a task is ill-defined or absent, their applicability becomes severely limited. To overcome this limitation, there is an urgent need to propose a more scalable OOD detection method that is independent of both pretext tasks and label supervision. We harness a new phenomenon called **Feature Resonance**, focusing on the feature space rather than the label space. We observe that, ideally, during the optimization of known ID samples, unknown ID samples undergo more significant representation changes than OOD samples, even when the model is trained to align arbitrary targets. The rationale behind it is that even without gold labels, the local manifold may still exhibit smooth resonance. Based on this, we further develop a novel graph OOD framework, dubbed **R**esonance-based **S**eparation and **L**earning (**RSL**), which comprises two core modules: (i)-a more practical micro-level proxy of feature resonance that measures the movement of feature vectors in one training step. (ii)-integrate with a synthetic OOD node strategy to train an effective OOD classifier. Theoretically, we derive an error bound showing the superior separability of OOD nodes during the resonance period. Extensive experiments on a total of thirteen real-world graph datasets empirically demonstrate that RSL achieves state-of-the-art performance. The code is available via https://github.com/ShenzhiYang2000/RSL.

## 1   Introduction

Graph-based machine learning models like Graph Neural Networks (GNNs) [Kipf and Welling, 2016a, Xu et al., 2018, Abu-El-Haija et al., 2019, Zhou et al., 2024] have become increasingly prevalent in applications such as social network analysis [Fan et al., 2019], knowledge graphs [Baek et al., 2020], and biological networks [De Cao and Kipf, 2018]. Despite the success of GNNs, detecting out-of-distribution (OOD) nodes remains an under-explored challenge. These OOD nodes differ significantly from the in-distribution (ID) nodes used during training, and their presence can severely undermine the performance and robustness of graph models. As deploying GNNs in real-world environments becomes more common, the ability to identify and handle OOD nodes is crucial for ensuring the reliability of using these models.

39th Conference on Neural Information Processing Systems (NeurIPS 2025).

To address this, most existing methods [Hendrycks and Gimpel, 2016, Liang et al., 2017, Hendrycks et al., 2018, Liu et al., 2020, Wu et al., 2023] employ classifiers pretrained on a preceding classification task to develop OOD metrics based on (i)-classifier outputs, such as Maximum Softmax Probability (MSP) [Hendrycks and Gimpel, 2016] and Energy [Liu et al., 2020, Wu et al., 2023]; (ii)-supervised representations, such as KNN [Sun et al., 2022] and NNGuide [Park et al., 2023]. These methods heavily rely on two key assumptions: (i) the availability of multi-class labels, and (ii) a well-defined pretext multi-class classification task. However, in practice, there exists a wide range of OOD detection scenarios that fall outside these constraints. In many cases, the pretext task is not classification—for example, OOD detection in generative modeling [Nalisnick et al., 2018, Ren et al., 2019], regression [Lakshminarayanan et al., 2017], or reinforcement learning [Nasvytis et al., 2024]. In some scenarios, there may not even be a defined pretext task at all, such as in one-class OOD detection [Ruff et al., 2018]. These non-classification settings lack accessible multi-class labels, making it difficult to directly apply existing methods. Therefore, there is an urgent need for label-agnostic and unsupervised approaches that can operate effectively in such contexts. To date, only a few papers [Gong and Sun, 2024, Sehwag et al., 2021, Liu et al., 2023] study this practical setup, and there is still a large room for improvement, especially in the graph field at the node level.

In this paper, we revisit the graph OOD detection task at the node level from a new perspective and turn our attention to the intrinsic similarities within the data. An intuitive idea is that the ID samples may still share some commonalities in the representation space. We hypothesize that when optimizing the representation of known ID nodes, the representation of unknown ID nodes and unknown OOD nodes will change with different trajectories. Based on the hypothesis and using a toy dataset (Figure 1(a)), we design an experiment where the features of labeled ID samples are aligned to an arbitrarily fixed representation vector. Interestingly, we observe a distinct behavior during this optimization process: the representations of unlabeled wild ID samples experienced more pronounced changes than wild OOD samples, as shown in Figure 1(b). This phenomenon closely resembles the concept of forced vibration, where resonance occurs when an external force aligns with the natural frequency of an oscillator, amplifying

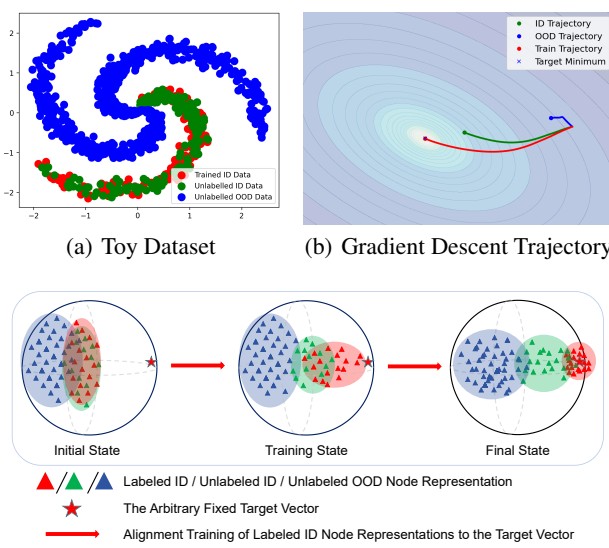

(a) Toy Dataset     (b) Gradient Descent Trajectory

(c) Schematic of Feature Resonance

Figure 1: (a) We conduct a preliminary study on the changes in ID and OOD node representations during training using a toy dataset. (b) Projections of the representations of ID and OOD nodes onto gradients: $\mathrm{Proj}_{\nabla\ell(\theta_t;\cdot)}\mathbf{x}_i = \frac{\mathbf{x}_i \cdot \nabla\ell(\theta_t;\cdot)}{\|\nabla\ell(\theta_t;\cdot)\|_2^2} \cdot \nabla\ell(\theta_t;\cdot)$. (c) Schematic of Feature Resonance.

its oscillation to a maximum. Analogously, we refer to this phenomenon as **Feature Resonance**: *during the optimization of known ID samples, the representation of unknown ID samples undergoes more significant changes compared to OOD samples.* This phenomenon reveals the intrinsic relationship between ID samples, highlighting their shared underlying distribution. Evidently, this feature resonance phenomenon can be leveraged for OOD detection: weaker representation changes during known ID optimization indicate a higher likelihood of being OOD.

In real-world scenarios, due to the intrinsic complex pattern in data, we find that the feature resonance phenomenon still occurs but slightly differs from the ideal conditions. To illustrate this, we further propose a micro-level proxy for measuring feature resonance—by computing the movement of the representation vector in one training step. Our findings reveal that in more complex scenarios, the feature resonance phenomenon typically arises during the middle stages of the training process, whereas during other phases, it may be overwhelmed by noise or obscured by overfitting. In such cases, evaluating the entire trajectory often fails to yield satisfactory results. Fortunately, efficient OOD detection can still be achieved by calculating the micro-level feature resonance measure.

By utilizing a simple binary ID/OOD validation set[1], we empirically show the feature resonance period can be precisely identified, and we identify more minor representation differences as OOD samples. Notably, our new micro-level feature resonance measure is still *label-independent* by fitting a randomly fixed target, making it highly compelling in category-free and task-agnostic scenarios. Theoretical and experimental proof that micro-level feature resonance can filter a set of reliable OOD nodes with low error. Furthermore, we combine the micro-level feature resonance with the current Langevin-based synthetic OOD nodes generating strategy to train an OOD classifier for more effective OOD node detection performance, which we call the whole framework as **RSL**; for example, the FPR95 metric is reduced by an average of **15.20**% compared to the current state-of-the-art methods.

## 2 Method

### 2.1 Revealing the Feature Resonance Phenomenon

Previous studies [Hendrycks and Gimpel, 2016, Liu et al., 2020, Wu et al., 2023] mostly train a classifier on ID nodes with multi-category labels and develop selection criteria based on output probabilities, e.g., entropy. However, these methods become inapplicable in category-free and task-agnostic scenarios.

To address this problem, we turn our attention to the intrinsic similarities within the data. An intuitive idea is that although the output space may no longer be reliable, the ID samples may still share some commonalities in the representation space. We hypothesize that when optimizing the representation of known ID nodes, the representation of unknown ID nodes and unknown OOD nodes will change with different trajectories. Motivated by this, and under the

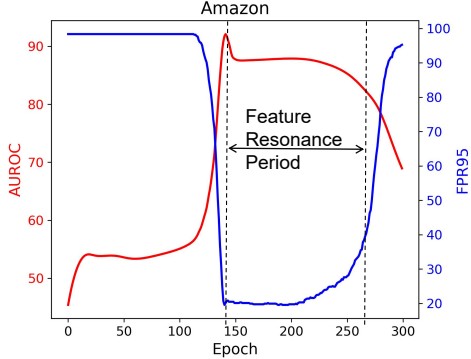

Figure 2: The performance of using resonance-based score $\tau$ to detect OOD nodes varies with training progress. The higher the AUROC, the better, and the lower the FPR95, the better.

assumption of some specific training process, we define a **feature trajectory measure** $\hat{F}(\tilde{\mathbf{x}}_i)$ of a sample $\tilde{\mathbf{x}}_i$:

$$\hat{F}(\tilde{\mathbf{x}}_i) = \sum_t h_{\theta_{t+1}}(\tilde{\mathbf{x}}_i) - h_{\theta_t}(\tilde{\mathbf{x}}_i) \tag{1}$$

where $h_{\theta_t}$ denote the model that performs representation transformation on a sample $\tilde{\mathbf{x}}_i$, with $\theta_t$ representing its parameters at the $t$-th epoch.

In our preliminary experiments, we first calculate the metric under *supervised conditions* and observe a significant difference between the feature trajectories of ID samples and those of OOD samples. Specifically, we perform multi-category training on known ID nodes on two datasets with true $N$-category labels, Squirrel and WikiCS [2]. Imagine that during multi-category training, representations of known ID nodes within the same category align while unknown ID nodes drift toward the corresponding category centers. However, the trajectory trends and lengths of unknown ID nodes differ significantly from those of OOD nodes, with the former showing more distinct trends and longer trajectories; see Figure 1 (c) for visual illustration. In other words, the well-defined in-distribution manifold is always shaped by ID samples, whose representation trajectories tend to exhibit similar behavior, which we refer to as feature resonance. Conversely, OOD samples belong to distinct manifold structures, making their representations less likely to converge coherently. Evidently, this feature resonance phenomenon can be leveraged for OOD detection.

Despite the promise, the abovementioned feature resonance phenomenon occurs under multi-category training. *But how can we induce this phenomenon in a label-agnostic scenario without multi-category labels?* Interestingly, we find that even when **random labels** are assigned to known ID nodes for multi-category training, the trajectories of unknown ID nodes are still more significant than

---

[1]The use of the validation set is consistent with previous works [Katz-Samuels et al., 2022, Gong and Sun, 2024, Du et al., 2024a,b] and does not contain multi-category labels.

[2]$N$ is the number of categories, and the results above with different target vectors are shown in Table 6.

those of unknown OOD nodes. More surprisingly, on a ideal toy dataset, even when all known ID node representations are aligned toward **one single random fixed target vector**, the trajectories of unknown ID nodes are still longer than those of unknown OOD nodes, as shown in Figure 1(a). Green points represent unknown ID samples, blue points represent unknown OOD samples, and red points represent known ID samples aligned to a target vector. As shown in Figure 1(b), modifying the representation of known ID samples results in longer representation change trajectories for unknown ID samples compared to unknown OOD samples. The experiments above indicate that the feature resonance phenomenon is *label-independent* and results from the intrinsic relationships between ID node representations. Therefore, this is highly suitable for category-free and task-agnostic OOD detection scenarios without multi-category labels.

Since the trajectory represents a global change, we call it a macroscopic feature resonance, as follows:

**Definition 1.** *Feature Resonance (macroscopic): For any optimization objective $\ell(\boldsymbol{X}_{known}, \cdot)$ applied to the representations $\boldsymbol{X}_{known}$ of known ID samples derived from any model $h_\theta(\cdot)$, we have $\parallel \hat{F}(\tilde{\mathbf{x}}_i) \parallel_{\mathbb{P}_{in}^{wild}} > \parallel \hat{F}(\tilde{\mathbf{x}}_i) \parallel_{\mathbb{P}_{out}^{wild}}$.*

### 2.2 Utilizing the Micro-level Feature Resonance Phenomenon with An Arbitrary Target

As mentioned above, we can leverage the feature resonance phenomenon to detect OOD nodes. In our realistic implementations, we align the features of known ID nodes to an arbitrary target vector using mean squared error as follows:

$$\ell(h_{\theta_t}(\boldsymbol{X}_{\text{known}}), e) = \mathbb{E}(\parallel \mathbf{1}^\top e - (\boldsymbol{X}_{\text{known}}\mathbf{W}^\top) \parallel_2^2) \tag{2}$$

where $h_{\theta_t}(\boldsymbol{X}_{\text{known}}) = \boldsymbol{X}_{\text{known}}\mathbf{W}^\top$ represent the last linear layer of the model for representation transformation and $e$ denotes an arbitrary randomly generated target vector.

But, in contrast to our toy dataset, the real-world datasets typically exhibit much more complex feature attributes. As a result, the feature resonance of trajectory at the macro level is not as ideal or pronounced as observed in experiments on the toy dataset. Therefore, to explore the reasons behind this issue, we delve deeper into the changes in finer-grained node representations across epochs to study the feature resonance phenomenon. Specifically, we study the differences in $\Delta h_{\theta_t}(\tilde{\mathbf{x}}_i) = h_{\theta_{t+1}}(\tilde{\mathbf{x}}_i) - h_{\theta_t}(\tilde{\mathbf{x}}_i)$ between ID samples and OOD samples. Obviously, the existence of $\parallel \Delta h_{\theta_t}(\tilde{\mathbf{x}}_i) \parallel_{\mathbb{P}_{in}^{wild}} > \parallel \Delta h_{\theta_t}(\tilde{\mathbf{x}}_i) \parallel_{\mathbb{P}_{out}^{wild}}$ is a necessary condition for satisfying $\parallel \hat{F}(\tilde{\mathbf{x}}_i) \parallel_{\mathbb{P}_{in}^{wild}} > \parallel \hat{F}(\tilde{\mathbf{x}}_i) \parallel_{\mathbb{P}_{out}^{wild}}$, so we define $\parallel \Delta h_{\theta_t}(\tilde{\mathbf{x}}_i) \parallel_{\mathbb{P}_{in}^{wild}} > \parallel \Delta h_{\theta_t}(\tilde{\mathbf{x}}_i) \parallel_{\mathbb{P}_{out}^{wild}}$ as a feature resonance at the microscopic level:

**Definition 2.** *Feature Resonance (microscopic): For any optimization objective $\ell(\boldsymbol{X}_{known}, \cdot)$ applied to the known ID nodes' representations $\boldsymbol{X}_{known}$ from any model $h_{\theta_t}(\cdot)$, during the optimization process, there exists $t$ such that $\parallel \Delta h_{\theta_t}(\tilde{\mathbf{x}}_i) \parallel_{\mathbb{P}_{in}^{wild}} > \parallel \Delta h_{\theta_t}(\tilde{\mathbf{x}}_i) \parallel_{\mathbb{P}_{out}^{wild}}$. We define the resonance-based filtering score as $\tau_i = \parallel \Delta h_{\theta_t}(\tilde{\mathbf{x}}_i) \parallel_2$. The resonance-based scores $\tau$ of OOD nodes should be smaller than those of ID nodes at $t$.*

By observing $\tau$ for ID samples and OOD samples, we find that feature resonance does not persist throughout the entire training process but rather occurs at specific stages of training. In our experiments on the common benchmarks, we find that during the early stages of training, the model is searching for the optimal optimization path, leading to chaotic representation changes and thus making feature resonance insignificant. However, in the middle stages of training, once the model identifies an optimization path that aligns with the patterns of the ID samples, it optimizes along the path most relevant to the features of the ID samples, and feature resonance becomes most prominent. As the model continues to optimize and enters the overfitting stage, the feature resonance phenomenon begins to dissipate. Figure 2 shows the experimental results on the Amazon dataset, and others are provided in Figure 3 of the Appendix. Through the above experiments and analyses, we find that using $\hat{F}(\tilde{\mathbf{x}}_i)$ to identify OOD nodes is affected by error accumulation and is, therefore, not a reliable approach. However, there exists a specific period during training when micro-level feature resonance occurs. By utilizing a validation set [Katz-Samuels et al., 2022, Gong and Sun, 2024, Du et al., 2024a,b], we can easily identify the period during which feature resonance occurs.

Formally, our new feature resonance-based OOD nodes detector is defined as follows:

$$g_\gamma(\tilde{\mathbf{x}}_i) = \mathbb{1}\{\tau_i^* \le \gamma\}, \quad \text{s.t., } \tau^* = \max_t \text{AUROC}(\tau_{\mathcal{V}_{\text{val}}^{\text{in}}}^t, \tau_{\mathcal{V}_{\text{val}}^{\text{out}}}^t) \tag{3}$$

Table 1: The statistics of the real-world OOD node detection datasets. $\times$ denotes no available multi-category labels. Notably, we do not use any true labels for all datasets.

| Dataset | Squirrel | WikiCS | Cora | Citeseer | Pubmed | Chameleon | YelpChi | Amazon | Reddit |
|---|---|---|---|---|---|---|---|---|---|
| # Nodes | 5,201 | 11,701 | 2,708 | 3,327 | 19,717 | 2,277 | 45,954 | 11,944 | 10,984 |
| # Features | 2,089 | 300 | 1,433 | 3,703 | 500 | 2,325 | 32 | 25 | 64 |
| Avg. Degree | 41.7 | 36.9 | 7.8 | 5.5 | 9.0 | 31.7 | 175.2 | 800.2 | 15.3 |
| OOD node (%) | 20.0 | 29.5 | 66.7 | 45.7 | 20.8 | 40.2 | 14.5 | 9.5 | 3.3 |
| # Category | 5 | 10 | 3 | 3 | 2 | 3 | $\times$ | $\times$ | $\times$ |

where $g_\gamma = 1$ indicates the OOD nodes while $g_\gamma = 0$ indicates otherwise, and $\gamma$ is typically chosen to guarantee a high percentage, such as 95%, of ID data that is correctly classified. Here, $t$ is determined by the validation set $\mathcal{V}_{\text{val}}$.

To summarize our method: we calculate a resonance-based filtering score $\tau$ during the transformation of known ID sample representations. By leveraging a validation set, we identify the period during training when micro-level resonance is most significant. Within this period, test set nodes with smaller $\tau$ values are more likely to be OOD nodes.

## 2.3 Extension with Synthetic OOD Node Strategy

Although the resonance-based filtering score effectively separates OOD nodes, recent studies [Gong and Sun, 2024] suggest that training an OOD classifier with synthetic OOD nodes can improve OOD node detection. Therefore, we propose a novel framework that employs feature resonance scores to generate more realistic synthetic OOD nodes.

Specifically, we define the candidate OOD node set as $\mathcal{V}_{\text{cand}} = \{\tilde{v}_i \in \mathcal{V}_{\text{wild}} : \tau_i \leq T\}$, where $T = \min_n(\tau)$ is the $n$-th smallest $\tau$ of wild nodes, selecting nodes with the smallest $n$ $\tau$ values. The features of these nodes form $\boldsymbol{X}_{\text{cand}}$. Then, we compute a trainable metric based on the weighted mapping of node $v$'s representations across $K$ GNN layers: $E_\theta(v) = \mathbf{W}_K\big(\sum_k^K \beta_k \mathbf{h}_v^{(k)}\big)$, where $\beta_k \in \mathbb{R}$ is a learnable parameter, and $\mathbf{W}_K \in \mathbb{R}^{1 \times d}$ transforms the node representations to the energy scalar. Then, we employ stochastic gradient Langevin dynamics (SGLD) [Welling and Teh, 2011] to generate synthetic OOD nodes $\mathcal{V}_{\text{syn}} = \{\hat{v}_1, \cdots, \hat{v}_j\}$ with random initial features $\boldsymbol{X}_{\text{syn}} = \{\hat{\mathbf{x}}_1, \cdots, \hat{\mathbf{x}}_j\}$ as follows:

$$\hat{\mathbf{x}}_j^{(t+1)} = \lambda\big(\hat{\mathbf{x}}_j^{(t)} - \frac{\alpha}{2}\nabla_{\hat{\mathbf{x}}_j^{(t)}} E_\theta\big(\hat{v}_j^{(t)}\big) + \epsilon\big) + (1-\lambda)\mathbb{E}_{\mathbf{x} \sim \boldsymbol{X}_{\text{cand}}}\big(\mathbf{x} - \hat{\mathbf{x}}_j^{(t)}\big) \tag{4}$$

where $\frac{\alpha}{2}$ is the step size and $\lambda$ is a trade-off hyperparameter. $\epsilon$ is the Gaussian noise sampled from multivariate Gaussian distribution $\mathcal{N}(0, \zeta)$. Unlike Energy*Def* [Gong and Sun, 2024], we utilize the candidate OOD nodes $\mathcal{V}_{\text{cand}}$ as examples to generate synthetic OOD nodes that better align with the actual OOD nodes. After obtaining the synthetic OOD nodes, we define the training set $\mathcal{V}_{\text{train}} = \mathcal{V}_{\text{known}} \cup \mathcal{V}_{\text{cand}} \cup \mathcal{V}_{\text{syn}}$ with features $\boldsymbol{X}_{\text{train}}$ and labels $\boldsymbol{Y}_{\text{train}}$. The initially known ID nodes $\mathcal{V}_{\text{known}}$ are assigned a label of $1$. In contrast, the candidate OOD nodes $\mathcal{V}_{\text{cand}}$ and the generated synthetic OOD nodes $\mathcal{V}_{\text{syn}}$ are assigned a label of $0$. We use binary cross-entropy loss for training:

$$\ell_{\text{cls}} = -\big(\mathrm{y}_v\log(\sigma(E_\theta(v))) + (1 - \mathrm{y}_v)\log(1 - \sigma(E_\theta(v)))\big) \tag{5}$$

where $\sigma(\cdot)$ is the sigmod function. Similarly, we identify the OOD nodes as follows: $g'_{\gamma'}(E_\theta(v)) = \mathbb{1}\{E_\theta(v) \leq \gamma'\}$. , where $g'_{\gamma'} = 1$ indicates the OOD nodes while $g'_{\gamma'} = 0$ indicates otherwise, and $\gamma'$ is chosen to guarantee a high percentage, e.g., 95%, of ID data that is correctly classified.

## 2.4 Theoretical Analysis

Our main theorem quantifies the separability of the outliers in the wild by using the resonance-based filter score $\tau$. We provide detailed theoretical proof in the Appendix C.

Let $\text{ERR}_{\text{out}}^t$ be the error rate of OOD data being regarded as ID at $t$-th epoch, i.e., $\text{ERR}_{\text{out}}^t = |\{\tilde{v}_i \in \mathcal{V}_{\text{wild}}^{\text{out}} : \tau_i \geq T\}|/|\mathcal{V}_{\text{wild}}^{\text{out}}|$, where $\mathcal{V}_{\text{wild}}^{\text{out}}$ denotes the set of outliers from the wild data $\mathcal{V}_{\text{wild}}$. Then $\text{ERR}_{\text{out}}$ has the following generalization bound:

**Theorem 1.** *(Informal). Under mild conditions, if $\ell(\mathbf{x}, e)$ is $\beta$-smooth w.r.t $\mathbf{w}_t$, $\mathbb{P}_{\text{wild}}$ has $(\gamma, \xi)$-discrepancy w.r.t $\mathbb{P}_{\text{in}}$, and there is $\eta \in (0, 1)$ s.t. $\Delta = (1 - \eta)^2 \xi^2 - 8\beta_1 R_{in}^* > 0$, then where*

$n = \Omega(d/\min\{\eta^2\Delta, (\gamma - R_{in}^*)\}), m = \Omega(d/\eta^2\xi^2)$, *with the probability at least 0.9, for $0 < T <$* $0.9\widehat{M}_t(\widehat{M}_t$ *is the upper bound of score $\tau_i$),*

$$ERR_{out}^t \leq \frac{\max\{0, 1 - \Delta_\xi^\eta/\pi\}}{1 - T/(\sqrt{2}/(2t\alpha - 1))^2} + O(\sqrt{\frac{d}{\pi^2 n}}) + O(\sqrt{\frac{\max\{d, \Delta_\xi^{\eta^2}/\pi^2\}}{\pi^2(1 - \pi)m}}) \qquad (6)$$

*where $\Delta_\xi^\eta = 0.98\eta^2\xi^2 - 8\beta_1 R_{in}^*$ and $R_{in}^*$ is the optimal ID risk, i.e., $R_{in}^* = \min_{\mathbf{w} \in \mathcal{W}} \mathbb{E}_{\mathbf{x} \sim \mathbb{P}_{in}} \ell(\mathbf{x}, e)$. $d$ is the dimension of the space $\mathcal{W}$, $t$ denotes the $t$-th epoch, and $\pi$ is the OOD class-prior probability in the wild.*

**Practical implications of Therorem 1.** The above theorem states that under mild assumptions, the error $ERR_{out}$ is upper bounded. If the following two regulatory conditions hold: 1) the sizes of the labeled ID $n$ and wild data $m$ are sufficiently large; 2) the optimal ID risk $R_{in}^*$ is small, then the upper bound is mainly depended on $T$ and $t$. We further study the main error of $T$ and $t$ which we defined as $\delta(T, t)$.

**Theorem 2.** *(Informal). 1) if $\Delta_\xi^\eta \geq (1 - \epsilon)\pi$ for a small error $\epsilon \geq 0$, then the main error $\delta(T, t)$ satisfies that*

$$\delta(T, t) = \frac{\max\{0, 1 - \Delta_\xi^\eta/\pi\}}{1 - T/(\sqrt{2}/(2t\alpha - 1))^2} \leq \frac{\epsilon}{1 - T/(\sqrt{2}/(2t\alpha - 1))^2} \qquad (7)$$

*2) When learning rate $\alpha$ is small sufficiently, and if $\xi \geq 2.011\sqrt{8\beta_1 R_{in}^* + 1.011\sqrt{\pi}}$, then there exists $\eta \in (0, 1)$ ensuring that $\Delta > 0$ and $\Delta_\xi^\eta > \pi$ hold, which implies that the main error $\delta(T, t) = 0$.*

**Practical implications of Therorem 2.** Theorem 2 states that when the learning rate $\alpha$ is sufficiently small, the primary error $\delta(T, t)$ can approach zero if the difference $\zeta$ between the two data distributions $\mathbb{P}_{wild}$ and $\mathbb{P}_{in}$ is greater than a certain small value. Meanwhile, Theorem 2 also shows that the primary error $\delta(T, t)$ is inversely proportional to the learning rate $\alpha$ and the number of epochs ($t$). As the $t$ increases, the primary error $\delta(T, t)$ also increases, while a smaller learning rate $\alpha$ leads to a minor primary error $\delta(T, t)$. However, during training, there exists $t$ at which the error reaches its minimum.

## 3 Experiment

In this section, we present the main experimental results, while in **Appendix F.2**, we investigate feature resonance across datasets. **Appendix F.3** compares scoring methods, while **F.4** evaluates time efficiency. **Appendix F.5** tests RSL with different GNN encoders, and **F.6** examines graph-level OOD detection. **Appendices F.8** and **F.9** provide score distribution and node representation visualizations.

### 3.1 Experimental Setup

**Datasets.** We conduct extensive experiments to evaluate RSL on a total of nine real-world OOD node detection datasets: six multi-category datasets, Squirrel [Rozemberczki et al., 2021], WikiCS [Mernyei and Cangea, 2020], Cora, Citeseer, Pubmed [Kipf and Welling, 2016a], and Chameleon [Rozemberczki et al., 2021] and three binary classification fraud detection datasets: YelpChi [Rayana and Akoglu, 2015], Amazon [McAuley and Leskovec, 2013], and Reddit [Kumar et al., 2019]. The statistics of these datasets are summarized in Table 1. Additionally, we validate our method on four graph-level OOD detection datasets, including ENZYMES, PROTEINS [Morris et al., 2020], ClinTox, and LIPO [Wu et al., 2018]. We provide detailed dataset description in the Appendix E.3.

**Baselines.** We assess the performance of RSL against a total of twenty-one baseline methods spanning five categories: **1) Traditional outlier detection methods**, including local outlier factor [Breunig et al., 2000] like LOF-KNN and MLPAE. **2) Graph-based outlier detection models**, including GCN autoencoder [Kipf and Welling, 2016b], GAAN [Chen et al., 2020], DOMINANT [Ding et al., 2019], ANOMALOUS [Peng et al., 2018], and SL-GAD [Zheng et al., 2021]. **3) Transformation-based outlier detection approaches**, such as GOAD [Bergman and Hoshen, 2020] and NeuTral AD [Qiu et al., 2021]. **4) Entropy-based detection techniques**, including MSP, ODIN, OE, Energy, GKDE [Zhao et al., 2020], OODGAT [Song and Wang, 2022], GNNSafe [Wu et al.,

Table 2: Unsupervised OOD detection on real-world datasets. "OOM" indicates out-of-memory, "TLE" means time limit exceeded, and "-" denotes inapplicability. Detectors with ♣ use only node attributes, while ♠ share RSL's GNN backbone ( GCN ). Entropy-based methods with ◇ use true multi-category labels, and ♦ rely on K-means pseudo labels. Top results: **1st**, **2nd**.

| Dataset | Squirrel | | | WikiCS | | | YelpChi | | | Amazon | | | Reddit | | |
|---|---|---|---|---|---|---|---|---|---|---|---|---|---|---|---|
| **Method** | | | | | | | AUROC ↑ | AUPR ↑ | FPR@95 ↓ | | | | | | |
| LOF-KNN♣ | 51.85 | 29.87 | 95.21 | 44.06 | 37.48 | 96.28 | 56.39 | 25.98 | 92.57 | 45.25 | 14.26 | 95.10 | 57.88 | 6.95 | 93.24 |
| MLPAE♣ | 43.15 | 24.81 | 97.98 | 70.99 | 63.74 | 77.76 | 51.90 | 24.53 | 92.42 | 74.54 | 51.59 | 57.93 | 52.10 | 5.80 | 94.43 |
| GCNAE | 37.87 | 22.64 | 99.08 | 57.95 | 46.32 | 92.97 | 44.20 | 19.22 | 97.06 | 45.07 | 12.38 | 98.54 | 51.78 | 6.14 | 93.75 |
| GAAN | 38.01 | 22.57 | 98.99 | 58.15 | 46.60 | 93.37 | 44.29 | 19.30 | 96.91 | 53.26 | 6.63 | 98.05 | 52.21 | 5.96 | 94.06 |
| DOMINANT | 41.78 | 24.73 | 95.53 | 42.55 | 35.43 | 97.22 | 52.77 | 24.90 | 92.86 | 78.08 | 35.96 | 76.05 | 55.89 | 6.03 | 96.48 |
| ANOMALOUS | 51.04 | 29.09 | 96.39 | 67.99 | 54.51 | 92.74 | OOM | OOM | OOM | 65.12 | 25.15 | 85.34 | 55.18 | 6.40 | 94.10 |
| SL-GAD | 48.29 | 27.62 | 97.19 | 51.87 | 44.83 | 95.26 | 56.11 | 26.49 | 93.27 | 82.63 | 56.27 | 51.36 | 51.63 | 6.02 | 94.27 |
| GOAD♠ | 62.32 | 37.51 | 92.28 | 50.65 | 37.22 | 99.78 | 58.03 | 28.51 | 89.84 | 72.92 | 45.53 | 66.36 | 52.89 | 5.36 | 94.26 |
| NeuTral AD♠ | 52.51 | 30.04 | 97.16 | 53.58 | 43.49 | 94.30 | 55.81 | 25.14 | 94.23 | 70.01 | 24.36 | 92.19 | 55.70 | 6.45 | 94.59 |
| GKDE◇ | 56.15 | 33.41 | 94.96 | 70.47 | 61.18 | 82.71 | - | - | - | - | - | - | - | - | - |
| OODGAT◇ | 58.84 | 35.13 | 93.31 | 74.13 | 62.47 | 84.48 | - | - | - | - | - | - | - | - | - |
| GNNSafe♠◇ | 56.38 | 32.22 | 95.17 | 73.35 | 66.47 | 76.24 | - | - | - | - | - | - | - | - | - |
| NodeSafe♠◇ | 57.82 | 33.57 | 93.64 | 74.81 | 67.93 | 74.85 | - | - | - | - | - | - | - | - | - |
| GRASP♠◇ | 61.38 | 36.95 | 90.77 | 78.46 | 71.52 | 71.08 | - | - | - | - | - | - | - | - | - |
| OODGAT♦ | 57.78 | 34.66 | 92.61 | 52.76 | 44.71 | 90.02 | 55.97 | 23.07 | 97.93 | 82.54 | 54.94 | 52.10 | 54.62 | 6.05 | 93.85 |
| GNNSafe♠♦ | 49.52 | 26.63 | 97.60 | 64.15 | 50.85 | 92.63 | 55.26 | 26.68 | 91.40 | 68.51 | 25.39 | 84.31 | 49.63 | 5.36 | 95.98 |
| NodeSafe♠♦ | 50.91 | 27.48 | 96.18 | 65.77 | 52.02 | 91.03 | 55.61 | 28.01 | 89.95 | 69.92 | 26.44 | 82.72 | 50.74 | 6.03 | 94.26 |
| GRASP♠♦ | 52.63 | 28.12 | 94.87 | 66.94 | 53.33 | 89.12 | 58.05 | 28.67 | 88.11 | 70.31 | 27.81 | 81.29 | 51.82 | 6.91 | 92.04 |
| SSD♠ | TLE | TLE | TLE | 64.29 | 58.45 | 87.12 | 55.39 | 27.88 | 91.63 | 72.49 | 41.82 | 84.27 | 59.74 | 6.21 | 91.15 |
| EnergyDef♠ | 64.15 | 37.40 | 91.77 | 70.22 | 60.10 | 83.17 | 62.04 | 29.71 | 90.62 | 86.57 | 74.50 | 32.43 | 63.32 | 8.34 | 89.34 |
| RSL w/o classifier | 61.52 | 38.96 | 90.18 | 79.15 | 78.65 | 70.38 | 65.42 | 37.08 | 83.53 | 87.43 | 83.31 | 19.56 | 62.37 | 6.97 | 91.39 |
| RSL w/o $\mathcal{V}_{syn}$ | 60.46 | 34.89 | 93.59 | 81.21 | 79.93 | 52.19 | 65.15 | 38.93 | 81.84 | 87.81 | 81.10 | 25.18 | 61.36 | 8.48 | 89.43 |
| RSL | 64.12 | 39.58 | 89.90 | 84.01 | 81.14 | 49.23 | 66.11 | 39.73 | 80.45 | 90.03 | 83.91 | 19.60 | 64.83 | 10.18 | 85.49 |

2023], NodeSafe [Yang et al., 2025], and GRASP [Ma et al., 2024]. **5) Category-free detection methods**, including Energy*Def* [Gong and Sun, 2024] and SSD [Sehwag et al., 2021].

Additionally, we also compare our method with graph-level approaches, including **1) graph kernel combined with a detector** [Vishwanathan et al., 2010, Shervashidze et al., 2011, Neumann et al., 2016, Breunig et al., 2000, Manevitz and Yousef, 2001, Liu et al., 2008], **2) graph contrastive learning with a detector** [You et al., 2020, Liu et al., 2008, Sehwag et al., 2021, Zhou et al., 2021, Liu et al., 2023], and **3) end-to-end methods** [Zhao and Akoglu, 2023, Ma et al., 2022].

Details of baselines and implementation are in Appendix E.4 and E.5, respectively.

**Metrics.** Following prior research on OOD node detection, we evaluate the detection performance using three widely recognized, threshold-independent metrics: AUROC (↑), AUPR (↑) and FPR95(↓). We provide a detailed metric description in the Appendix E.2.

## 3.2 Main Results

Tables 2 and 3 present the main experimental results of various methods across nine public datasets. Specifically, when multi-class labels are unavailable, RSL significantly outperforms existing methods. For methods that require multi-class labels, we follow Energy*Def* [Gong and Sun, 2024] by assigning pseudo-labels using K-means. On the YelpChi, Amazon, and Reddit datasets, RSL achieves average improvements of 3.01%, 7.09%, and 8.95% over the SOTA methods in terms of AUROC, AUPR, and FPR95, respectively.

Table 3: Performance comparison across methods on Cora, Citeseer, Pubmed, and Chameleon.

| Dataset | Cora | | Citeseer | | Pubmed | | Chameleon | |
|---|---|---|---|---|---|---|---|---|
| **Method** | FPR@95 ↓ | | | AUROC ↑ | | | | |
| MSP | 70.86 | 84.56 | 67.81 | 82.39 | 87.37 | 68.80 | 85.70 | 57.96 |
| Energy | 67.54 | 85.47 | 88.53 | 72.38 | 93.86 | 54.09 | 88.06 | 59.20 |
| KNN | 90.20 | 70.94 | 83.10 | 72.91 | 89.79 | 64.14 | 93.38 | 57.90 |
| ODIN | 68.41 | 84.98 | 67.91 | 82.42 | 87.49 | 68.80 | 85.31 | 57.94 |
| Mahalanobis | 69.68 | 85.48 | 99.12 | 54.62 | 96.81 | 56.85 | 95.55 | 53.19 |
| GKDE | 63.71 | 86.27 | 80.42 | 79.94 | 65.48 | 69.92 | 92.93 | 50.14 |
| GPN | 58.45 | 82.93 | 65.68 | 88.13 | 88.61 | 64.13 | 82.25 | 68.20 |
| OODGAT | 94.59 | 53.63 | 62.39 | 84.33 | 88.27 | 58.28 | 94.43 | 59.67 |
| GNNSafe | 54.71 | 87.52 | 60.15 | 84.85 | 62.47 | 83.70 | 100.00 | 50.42 |
| NodeSafe | 50.32 | 89.11 | 55.71 | 86.16 | 58.07 | 85.11 | 98.76 | 52.19 |
| GRASP | 29.70 | 93.50 | 35.23 | 89.75 | 37.41 | 88.43 | 66.88 | 76.93 |
| RSL | 28.76 | 94.14 | 33.67 | 90.44 | 35.15 | 89.10 | 45.81 | 78.04 |

When multi-class label information is available, RSL shows even more significant performance gains on heterophilic graphs. On the Squirrel, WikiCS, and Chameleon datasets, RSL achieves an average improvement of **14.93%** in FPR95 over SOTA methods. This is because RSL does not rely on the homophily assumption of the graph, and thus performs well on heterophilic graphs. On homophilic graphs with multi-class labels—namely Cora, Citeseer, and Pubmed—RSL achieves performance comparable to SOTA methods. *Notably, RSL does not leverage multi-class labels for training in any*

Table 4: Performance of RSL at achievable label proportion $R$ in the WikiCS dataset.

| | $R = 0.0$ | $R = 0.1$ | $R = 0.2$ | $R = 0.4$ | $R = 0.6$ | $R = 0.8$ | $R = 1.0$ |
|---|---|---|---|---|---|---|---|
| AUROC($\uparrow$) | 84.01 | 86.41 | 87.46 | 87.89 | 88.35 | 88.57 | 89.07 |
| AUPR($\uparrow$) | 81.14 | 83.35 | 84.80 | 85.24 | 85.42 | 86.16 | 86.85 |
| FPR@95($\downarrow$) | 49.23 | 44.95 | 42.69 | 41.25 | 39.50 | 39.68 | 38.78 |

Table 5: Performance of different models on the Citeseer under varying homophily ratio $R$ values.

| Model | $R = 0.0$ | | $R = 0.1$ | | $R = 0.2$ | | $R = 0.3$ | | $R = 0.4$ | | $R = 0.5$ | |
|---|---|---|---|---|---|---|---|---|---|---|---|---|
| | FPR@95 $\downarrow$ | AUROC $\uparrow$ | | | | | | | | | | |
| GNNSafe | 60.15 | 84.85 | 52.92 | 86.56 | 47.68 | 87.40 | 45.64 | 89.30 | 42.22 | 90.42 | 37.19 | 91.78 |
| NodeSafe | 55.71 | 86.16 | 49.47 | 87.45 | 42.23 | 90.47 | 38.62 | 92.31 | 34.79 | 93.68 | 30.57 | 94.93 |
| GRASP | 35.23 | 89.75 | 30.63 | 91.26 | 27.35 | 94.00 | 16.01 | 96.36 | 13.69 | 97.33 | 10.08 | 98.25 |
| **RSL** | 33.67 | 90.44 | 28.91 | 91.75 | 16.62 | 94.45 | 9.82 | 96.09 | 7.71 | 97.30 | 4.53 | 98.23 |

*of the above experiments.* This highlights its label-agnostic and task-independent nature, contributing to its broader applicability.

**How effective is resonance-based filter score $\tau$?** The experimental results in the row labeled "RSL w/o classifier" of Table 2 show that using the raw resonance-based score $\tau$ to filter OOD nodes is already more effective than the SOTA method on most datasets. On the FPR95 metric, the score $\tau$ achieves an average reduction of 9.70% compared to current SOTA methods in Table 2.

**How effective are the synthetic OOD nodes combined with the feature resonance score?** The experimental results in the row labeled "RSL w/o $\mathcal{V}_{\text{syn}}$" of Table 2 show that after removing the synthetic OOD nodes, the performance of the trained OOD classifier declined to varying degrees. This indicates that synthetic OOD nodes enhance the generalization ability of the OOD classifier, allowing it to detect more OOD nodes more accurately. It is worth noting that our synthetic OOD nodes, generated by leveraging real OOD nodes selected using $\tau$, better align with real-world OOD scenarios and, therefore, outperform Energy*Def*.

**Can label information bring gains to RSL?** Although RSL can perform well in scenarios without multi-class labels, we want to investigate whether multi-class labels can bring similar benefits to RSL as they do for other methods. On the WikiCS dataset, we first pre-train the representations of the training set's ID nodes using supervised contrastive learning loss [Khosla et al., 2020] with training set labels at different proportions $R$, and then apply RSL. When $R = 1.0$, it indicates that RSL, like other methods that strictly require labels, uses all the training set labels. The results

Table 6: The effectiveness of the resonance-based filter score $\tau$ in filtering OOD nodes with different alignment targets for known ID node representations. **True multi-label** means aligning ID node representations with multiple target vectors based on true multi-class labels. **Multiple random vectors** means aligning ID node representations with random target vectors. **A random vector** means aligning ID node representations with a single target vector.

| Dataset Method | Target | Squirrel | | | WikiCS | | |
|---|---|---|---|---|---|---|---|
| | | AUROC $\uparrow$ | AUPR $\uparrow$ | FPR@95 $\downarrow$ | AUROC $\uparrow$ | AUPR $\uparrow$ | FPR@95 $\downarrow$ |
| Energy*Def* | - | 64.15 | 37.40 | 91.77 | 70.22 | 60.10 | 83.17 |
| RSL w/o classifier | True multi-label | 61.63 | 37.12 | 90.62 | 71.03 | 72.47 | 81.96 |
| RSL w/o classifier | Multiple random vectors | 61.44 | 37.39 | 90.62 | 73.64 | 74.13 | 69.25 |
| RSL w/o classifier | A random vector | 61.52 | 38.96 | 90.18 | 79.15 | 78.65 | 70.38 |

in Table 4 show that as the available label proportion increases, the ID node representations of the training set are better initialized, and RSL performs better. We believe this is because when the ID node representations are well-initialized, feature resonance is more easily induced and is more pronounced.

**Can graph homophily bring gains to RSL?** Most existing OOD node detection methods benefit from graph homophily, so we aim to explore whether RSL can also gain from it. We conduct experiments on the Citeseer dataset under varying levels of homophily, by removing a proportion $R$ of heterophilous edges and adding the same proportion $R$ of homophilous edges. The results in Table 5 show that as graph homophily improves, the performance of RSL also improves. We believe this is because enhanced graph homophily leads to more consistent representations among ID nodes and more pronounced differences between ID and OOD node representations, thereby making feature resonance easier to induce and more strongly expressed.

**How does feature resonance occur due to different target vectors?** We explore micro-level feature resonance using different target vectors through experiments on Squirrel and WikiCS datasets with true $N$-category labels. Based on neural collapse theory [Papyan et al., 2020, Zhou et al.,

Table 7: The effectiveness of different OOD candidate node selection strategies.

| Dataset / Method | Squirrel | | | WikiCS | | | YelpChi | | | Amazon | | | Reddit | | |
|---|---|---|---|---|---|---|---|---|---|---|---|---|---|---|---|
| | | | | | | | AUROC ↑ | AUPR ↑ | FPR@95 ↓ | | | | | | |
| RSL w/ Cosine Similarity | 64.00 | 38.11 | 91.46 | 81.61 | 76.36 | 70.38 | 59.76 | 35.03 | 85.89 | 83.35 | 74.85 | 27.63 | 54.07 | 7.25 | 92.21 |
| RSL w/ Euclidean Distance | 64.01 | 39.30 | 90.45 | 78.63 | 74.28 | 63.26 | 52.53 | 24.20 | 93.53 | 53.08 | 18.29 | 93.64 | 62.19 | 8.38 | 90.90 |
| RSL w/ Mahalanobis Distance | TLE | TLE | TLE | 83.18 | 79.11 | 58.03 | 54.07 | 25.44 | 92.40 | 63.71 | 30.66 | 79.96 | 60.81 | 8.42 | 90.08 |
| RSL w/ *EnergyDef* | 63.66 | 38.29 | 91.69 | 61.21 | 50.41 | 90.42 | 57.33 | 26.79 | 91.90 | 77.72 | 55.23 | 54.52 | 61.90 | 8.55 | 89.51 |
| RSL w/ Resonance-based Score $\tau$ | 64.12 | 39.58 | 89.90 | 84.01 | 81.14 | 49.23 | 66.11 | 39.73 | 80.45 | 90.03 | 83.91 | 19.60 | 64.83 | 10.18 | 85.49 |

2022], we set $N$ target vectors that form a simplex equiangular tight frame [3], maximizing separation. As shown in Table 6, the "True multi-label" row demonstrates the effectiveness of this approach. Interestingly, even when random labels are assigned (the "Multiple random vectors" row) or when all ID representations align with a fixed vector (the "A random vector" row), unknown ID nodes still show larger $\tau$ than unknown OOD nodes, as seen in Table 6. These results suggest that feature resonance is *label-independent*, stemming from intrinsic relationships between ID node representations.

**How do other OOD node selection methods perform?** We aim to evaluate the performance of RSL when integrated with methods other than the resonance-based score for selecting reliable OOD nodes. To ensure fairness, we used the same parameters and selected the same number of OOD nodes. From a metric learning perspective, we computed the cosine similarity, Euclidean distance, and Mahalanobis distance between unknown nodes and the prototypes of known ID nodes, with smaller values indicating a higher likelihood of being OOD nodes. We also applied Energy*Def* for OOD node selection. The results, presented in Table 7, show that, under the same conditions, the OOD nodes selected using $\tau$ are more reliable than those selected by the other methods.

# 4  Related Work

*General OOD Detection Methods.*  OOD detection methods fall into three main categories: **entropy-based**, **density-based**, and **representation-based** approaches. Entropy-based methods (e.g., MSP [Hendrycks and Gimpel, 2016], Energy [Liu et al., 2020], and others [Liang et al., 2017, Bendale and Boult, 2016, Hendrycks et al., 2018, Geifman and El-Yaniv, 2019, Malinin and Gales, 2018, Jeong and Kim, 2020, Chen et al., 2021, Wei et al., 2021, Ming et al., 2022b,a]) compute scores from class distributions but rely heavily on labeled data, making them less suitable for label-free scenarios. Density-based methods [Lee et al., 2018, Zisselman and Tamar, 2020] estimate sample likelihoods but struggle with high-dimensional, complex data [Ren et al., 2019, Serrà et al., 2019]. Representation-based methods like KNN [Sun et al., 2022] and NNGuide [Park et al., 2023] operate in embedding space but still require pre-trained ID classifiers. In contrast, SSD [Sehwag et al., 2021] avoids label dependence by using self-supervised learning on unlabeled ID data.

*General OOD Node Detection Methods.*  **Entropy-based methods**, such as MSP Hendrycks and Gimpel [2016], ODIN Liang et al. [2017], OE Hendrycks et al. [2018], Energy & Energy FineTune Liu et al. [2020], OODGAT [Song and Wang, 2022], GNNSafe [Wu et al., 2023], NodeSafe [Yang et al., 2025], and GRASP [Ma et al., 2024], as well as recent approaches like GOLD [Wang et al., 2025], EDBD [Um et al., 2025], and DeGEM [Chen et al., 2025], all rely on the outputs of a pre-trained classifier, making them unsuitable for unsupervised settings. **Graph anomaly detection methods**, like DOMINANT [Ding et al., 2019] and SL-GAD [Zheng et al., 2021], detect general anomalies through reconstruction errors, but they struggle to distinguish between OOD nodes and general anomalies.

*Unsupervised OOD Node Detection Methods.*  Unsupervised OOD node detection in graphs aims to identify OOD nodes without relying on multi-category labels and pretext classification tasks, posing unique challenges for traditional methods. Recent works [Li et al., 2022, Bazhenov et al., 2022, Liu et al., 2023, Ding and Shi, 2023] explore graph-level OOD detection but can not be directly applied to node-level OOD detection due to the complexity of node dependencies. Energy*Def* [Gong and Sun, 2024] employs a synthetic OOD node strategy for unsupervised OOD node detection, and we follow up by significantly improving OOD node detection performance in the unsupervised setting.

---

[3]The definition of the simplex equiangular tight frame is introduced in Appendix 6.

## 5 Conclusion

In this paper, we introduce the concept of **Feature Resonance** for unsupervised OOD node detection, demonstrating that unknown ID samples undergo more substantial representation changes compared to OOD samples during the optimization of known ID samples, even in the absence of multi-class labels. To effectively capture this phenomenon, we propose a label-independent, micro-level proxy that measures feature vector movements in a single training step. Building on this, we present the **RSL** framework, which integrates the micro-level feature resonance with synthetic OOD node generation via SGLD, enhancing OOD detection performance and offering an efficient and practical solution for unsupervised OOD node detection.

## 6 Acknowledgement

This paper is mainly supported by the NSFC under Grants (No. 62402424). Haobo Wang is also supported by the Fundamental Research Funds for the Central Universities (No. 226-2025-00085) and Zhejiang Provincial Universities (No. 226-2025-00065).

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

# A Notations, Definitions, Assumptions and Important Constants

## A.1 Notations

Table 8: Table of Notations and Descriptions

| Notation | Description |
|---|---|
| \multicolumn | Spaces |
| $X, Y$ | the input space and the label space. |
| $\mathcal{W}$ | the hypothesis spaces. |
| | Distributions |
| $\mathbb{P}_{\text{wild}}, \mathbb{P}_{\text{in}}, \mathbb{P}_{\text{out}}$ | data distribution for wild data, labeled ID data and OOD data. |
| $\mathbb{P}_{XY}$ | the joint data distribution for ID data.. |
| | Data and Models |
| $\mathbf{w}, \mathbf{x}$ | weight, input. |
| $\widehat{\nabla}, \tau$ | the average gradients on labeled ID data, uncertainty score. |
| $e$ | randomly generated unit vector. |
| $y$ | target unit vector $e$ for ID node representations. |
| $\widehat{y}_{\mathbf{x}}$ | predicted vector for input $\mathbf{x}$. |
| $h_{\theta_t}$ | predictor on labeled in-distribution |
| $X_{\text{wild}}^{\text{in}}, X_{\text{wild}}^{\text{out}}$ | inliers and outliers in the wild dataset. |
| $X^{\text{in}}, X_{\text{wild}}$ | labeled ID data and unlabeled wild data. |
| $n, m$ | size of $X^{\text{in}}$, size of $X_{\text{wild}}$ |
| $T$ | the filtering threshold |
| $X_T$ | wild data whose uncertainty score higher than threshold $T$ |
| | Distances |
| $r_1$ | the radius of the hypothesis spaces $\mathcal{W}$ |
| $\|\cdot\|_2$ | $\ell_2$ norm |
| | Loss, Risk and Predictor |
| $\ell(\cdot, \cdot)$ | ID loss function |
| $R_X(h_{\theta_t})$ | the empirical risk w.r.t. predictor $h_{\theta_t}$ over data $X$ |
| $R_{\mathbb{P}_{XY}}(h_{\theta_t})$ | the risk w.r.t. predictor $h_{\theta_t}$ over distribution $\mathbb{P}_{XY}$. |
| $ERR_{\text{out}}$ | the error rate of regarding OOD as ID. |

## A.2 Definitions

**Definition 3.** *($\beta$ -smooth). We say a loss function $\ell(h_{\theta_t}(\mathbf{x}), y)$ (defined over $X \times Y$) is $\beta$ -smooth, if for any $\mathbf{x} \in X$ and $y \in Y$*

$$\|\nabla \ell(h_{\theta_t}(\mathbf{x}), y) - \nabla \ell(h_{\theta_t}(\mathbf{x}), y)\|_2 \leq \beta \|\mathbf{w} - \mathbf{w}'\|_2$$

**Definition 4.** *(Gradient-based Distribution Discrepancy). Given distributions $\mathbb{P}$ and $\mathbb{Q}$ defined over $X$ , the Gradient-based Distribution Discrepancy w.r.t. predictor $\mathbf{f}_{\text{w}}$ and loss $t$ is*

$$d_{\mathbf{w}}^{\ell}(\mathbb{P}, \mathbb{Q}) = \left\|\nabla R_{\mathbb{P}}(h_{\theta_t}, \widehat{h}_{\theta}) - \nabla R_{\mathbb{Q}}(h_{\theta_t}, \widehat{h}_{\theta})\right\|_2,$$

*where $\widehat{h}_{\theta}$ is a classifier which returns the closest one-hot vector of $h_{\text{w}}$: $R_{\mathbb{P}}(h_{\theta_t}, \widehat{h}_{\theta}) = \mathbb{E}_{\mathbf{x} \sim \mathbb{P}} \ell(h_{\theta_t}, \widehat{h}_{\theta})$ and $R_{\mathbb{Q}}(h_{\theta_t}, \widehat{h}_{\theta}) = \mathbb{E}_{\mathbf{x} \sim \mathbb{Q}} \ell(h_{\theta_t}, \widehat{h}_{\theta})$*

**Definition 5.** *($\gamma, \xi$) -discrepancy). We say a wild distribution $\mathbb{P}_{wild}$ has $(\gamma, \xi)$ -discrepancy w.r.t. an ID joint distribution $\mathbb{P}_{in\ n}$, if $\gamma > \min_{\mathbf{w} \in \mathcal{W}} R_{\mathbb{P}_{XY}}(h_{\theta})$ and for any parameter $\mathbf{w} \in \mathcal{W}$ satisfying that $R_{\mathbb{P}, XY}(h_{\theta_t}) \leq \gamma$ should meet the following condition*

$$d_{\mathbf{w}}^{\ell}(\mathbb{P}_{in}, \mathbb{P}_{wild}) > \xi,$$

*where $R_{\mathbb{P}_{XY}}(h_{\theta}) = \mathbb{E}_{(\mathbf{x}, y) \sim \mathbb{P}_{XY}} \ell(h_{\theta}(\mathbf{x}), y)$*

### A.3 Assumptions

**Assumption 1.**

- The parameter space $\mathcal{W} \subset B(\mathbf{w}_0, r_1) \subset \mathbb{R}^d$ ($\ell_2$ ball of radius $r_1$ around $W_0$);
- $\ell(h_{\theta_t}(\mathbf{x}), y) \geq 0$ and $\ell(h_{\theta_t}(\mathbf{x}), y)$ is $\beta_1$-smooth;
- $\sup_{(\mathbf{x},y) \in \mathbf{X} \times \mathbf{Y}} \|\nabla \ell(h_{\theta_0}(\mathbf{x}), y)\|_2 = b_1$;
- $\sup_{(\mathbf{x},y) \in \mathbf{X} \times \mathbf{Y}} \ell(h_{\theta_0}(\mathbf{x}), y) = B_1$.

**Assumption 2.** $\ell(\mathbf{f}(\mathbf{x}), \widehat{y}_{\mathbf{x}}) \leq \min_{y \in \mathbf{Y}} \ell(\mathbf{f}(\mathbf{x}), y)$, where $\widehat{y}_{\mathbf{x}}$ returns the closest vector of the predictor $\mathbf{f}$'s output on $\mathbf{x}$

### A.4 Constants in Theory

Table 9: Constants in theory.

| Constants | Description |
|---|---|
| $M = \beta_1 r_1^2 + b_1 r_1 + B_1$ | the upper bound of loss $\ell(h_{\theta_t}(\mathbf{x}), y)$. |
| $M' = 2(\beta_1 r_1 + b_1)^2$ | the upper bound of gradient-based filtering score [Du et al., 2024a] |
| $\widehat{M}_t = (\sqrt{M'/2} + 1)/(2t)$ | the upper bound of our resonance-based filtering score $\tau$ at the $t$-th epoch |
| $\tilde{M} = \beta_1 M$ | a constant for simplified representation |
| $d$ | the dimensions of parameter spaces $\mathcal{W}$ |
| $R_{\text{in}}^*$ | the optimal ID risk, i.e., $R_{in}^* = \min_{\mathbf{w} \in \mathcal{W}} \mathbb{E}_{\mathbf{x} \sim \mathbb{P}_{\text{in}}} \mathcal{L}_1(\mathbf{x}, e)$ |
| $\delta(T, t)$ | the main error in Eq. 7 |
| $\xi$ | the discrepancy between $\mathbb{P}_{\text{in}}$ and $\mathbb{P}_{\text{wild}}$ |
| $\pi$ | the ratio of OOD distribution in $\mathbb{P}_{\text{wild}}$ |
| $\alpha$ | learning rate |

## B  Main Theorems

**Theorem 3.** *If Assumptions 1 and 2 hold, $\mathbb{P}_{wild}$ has $(\gamma, \xi)$-discrepancy w.r.t. $\mathbb{P}_{xy}$, and there exists $\eta \in (0, 1)$ s.t. $\Delta = (1 - \eta)^2 \xi^2 - 8\beta_1 R_{in}^* > 0$, then for*

$$n = \Omega\Big(\frac{\tilde{M} + M(r_1 + 1)d}{\eta^2 \Delta} + \frac{M^2 d}{(\gamma - R_{in}^*)^2}\Big), \quad m = \Omega\Big(\frac{\tilde{M} + M(r_1 + 1)d}{\eta^2 \xi^2}\Big),$$

*with the probability at least 9/10 for any $0 < T < \widehat{M}_t$ (here $\widehat{M}_t$ is the upper bound of filtering score $\tau_i$ at $t$-th epoch, i.e., $\tau_i \leq \widehat{M}_t$ )*

$$ERR_{out}^t \leq \frac{\max\{0, 1 - \Delta_\xi^\eta/\pi\}}{1 - T/(\sqrt{2}/(2t\alpha - 1))^2} + O\Big(\sqrt{\frac{d}{\pi^2 n}}\Big) + O\Big(\sqrt{\frac{\max\{d, \Delta_\xi^{\eta^2}/\pi^2\}}{\pi^2(1 - \pi)m}}\Big) \quad (8)$$

*where $\Delta_\xi^\eta = 0.98\eta^2 \xi^2 - 8\beta_1 R_{in}^*$ and $R_{in}^*$ is the optimal ID risk, i.e., $R_{in}^* = \min_{\mathbf{w} \in \mathcal{W}} \mathbb{E}_{\mathbf{x} \sim \mathbb{P}_{\text{in}}} \mathcal{L}_1(\mathbf{x}, e)$. $d$ is the dimension of the space $\mathcal{W}$, $t$ denotes the $t$-th epoch, and $\pi$ is the OOD class-prior probability in the wild.*

$$M = \beta_1 r_1^2 + b_1 r_1 + B_1, \quad \tilde{M} = M\beta_1 \quad (9)$$

**Theorem 4.** *1) if $\Delta_\xi^\eta \geq (1 - \epsilon)\pi$ for a small error $\epsilon \geq 0$, then the main error $\delta(T, t)$ satisfies that*

$$\delta(T, t) = \frac{\max\{0, 1 - \Delta_\xi^\eta/\pi\}}{1 - T/(\sqrt{2}/(2t\alpha - 1))^2} \leq \frac{\epsilon}{1 - T/(\sqrt{2}/(2t\alpha - 1))^2} \quad (10)$$

*2) When learning rate $\alpha$ is small sufficiently, and if $\xi \geq 2.011\sqrt{8\beta_1 R_{in}^*} + 1.011\sqrt{\pi}$, then there exists $\eta \in (0, 1)$ ensuring that $\Delta > 0$ and $\Delta_\xi^\eta > \pi$ hold, which implies that the main error $\delta(T, t) = 0$.*

## C Proofs of Main Theorems

### C.1 Proof of Theorem 1

Step 1. With the probability at least $1 - \frac{7}{3}\delta > 0$

$$
\begin{aligned}
\mathbb{E}_{\tilde{\mathbf{x}}_i \sim S^{\mathrm{in}}_{\mathrm{wild}}} \tau_i \leq{} & 8\beta_1 R^*_{\mathrm{in}} \\
& + 4\beta_1 \Big[ C\sqrt{\frac{Mr_1(\beta_1 r_1 + b_1)d}{n}} + C\sqrt{\frac{Mr_1(\beta_1 r_1 + b_1)d}{(1-\pi)m - \sqrt{m\log(6/\delta)/2}}} \\
& + 3M\sqrt{\frac{2\log(6/\delta)}{n}} + M\sqrt{\frac{2\log(6/\delta)}{(1-\pi)m - \sqrt{m\log(6/\delta)/2}}} \Big],
\end{aligned}
$$

This can be proven by Lemma 7 in [Du et al., 2024a] and following inequality

$$
\mathbb{E}_{\tilde{\mathbf{x}}_i \sim S^{in}_{wild}} \tau_i \geq \mathbb{E}_{\tilde{\mathbf{x}}_i \sim \boldsymbol{X}^m_{wild}} \left\| \nabla \ell(h_{\theta_{\boldsymbol{X}^m}}(\tilde{\mathbf{x}}_i), \widehat{h}_{\theta_{\boldsymbol{X}^m}}(\tilde{\mathbf{x}}_i)) - \mathbb{E}_{(\mathbf{x}_j, y_j) \sim \boldsymbol{X}^m} \nabla \ell(h_{\theta_{\boldsymbol{X}^m}}(\mathbf{x}_j), y_j) \right\|^2_2,
$$

Step 2. It is easy to check that

$$
\mathbb{E}_{\tilde{\mathbf{x}}_i \sim \boldsymbol{X}_{\mathrm{wild}}} \tau_i = \frac{|\boldsymbol{X}^{\mathrm{in}}_{\mathrm{wild}}|}{|\boldsymbol{X}_{\mathrm{wild}}|} \mathbb{E}_{\tilde{\mathbf{x}}_i \sim \boldsymbol{X}^{\mathrm{in}}_{\mathrm{wild}}} \tau_i + \frac{|\boldsymbol{X}^{\mathrm{out}}_{\mathrm{wild}}|}{|\boldsymbol{X}_{\mathrm{wild}}|} \mathbb{E}_{\tilde{\mathbf{x}}_i \sim \boldsymbol{X}^{\mathrm{out}}_{\mathrm{wild}}} \tau_i.
$$

Step 3. Let

$$
\begin{aligned}
\epsilon(n, m) ={} & 4\beta_1 \Big[ C\sqrt{\frac{Mr_1(\beta_1 r_1 + b_1)d}{n}} + C\sqrt{\frac{Mr_1(\beta_1 r_1 + b_1)d}{(1-\pi)m - \sqrt{m\log(6/\delta)/2}}} \\
& + 3M\sqrt{\frac{2\log(6/\delta)}{n}} + M\sqrt{\frac{2\log(6/\delta)}{(1-\pi)m - \sqrt{m\log(6/\delta)/2}}} \Big].
\end{aligned}
$$

Under the condition in Theorem 5 in [Du et al., 2024a], with the probability at least $\frac{97}{100} - \frac{7}{3}\delta > 0$

$$
\begin{aligned}
\mathbb{E}_{\tilde{\mathbf{x}}_i \sim \boldsymbol{X}^{\mathrm{out}}_{\mathrm{wild}}} \tau_i &\leq \frac{m}{|\boldsymbol{X}^{\mathrm{out}}_{\mathrm{wild}}|} \Big[ \frac{98\eta^2\xi^2}{100} - \frac{|\boldsymbol{X}^{\mathrm{in}}_{\mathrm{wild}}|}{m} 8\beta_1 R^*_{\mathrm{in}} - \frac{|\boldsymbol{X}^{\mathrm{in}}_{\mathrm{wild}}|}{m} \epsilon(n, m) \Big] \\
&\leq \frac{m}{|\boldsymbol{X}^{\mathrm{out}}_{\mathrm{wild}}|} \Big[ \frac{98\eta^2\xi^2}{100} - 8\beta_1 R^*_{\mathrm{in}} - \epsilon(n, m) \Big] \\
&\leq \Big[ \frac{1}{\pi} - \frac{\sqrt{\log 6/\delta}}{\pi^2\sqrt{2m} + \pi\sqrt{\log(6/\delta)}} \Big] \Big[ \frac{98\eta^2\xi^2}{100} - 8\beta_1 R^*_{\mathrm{in}} - \epsilon(n, m) \Big].
\end{aligned}
$$

In this proof, we set

$$
\Delta(n, m) = \Big[ \frac{1}{\pi} - \frac{\sqrt{\log 6/\delta}}{\pi^2\sqrt{2m} + \pi\sqrt{\log(6/\delta)}} \Big] \Big[ \frac{98\eta^2\xi^2}{100} - 8\beta_1 R^*_{\mathrm{in}} - \epsilon(n, m) \Big].
$$

Note that $\Delta^\eta_\xi = 0.98\eta^2\xi^2 - 8\beta_1 R^*_{\mathrm{in}}$ , then

$$
\Delta(n, m) = \frac{1}{\pi}\Delta^\eta_\xi - \frac{1}{\pi}\epsilon(n, m) - \Delta^\eta_\xi \epsilon(m) + \epsilon(n)\epsilon(n, m),
$$

where $\epsilon(m) = \sqrt{\log 6/\delta}/(\pi^2\sqrt{2m} + \pi\sqrt{\log(6/\delta)})$.

Step 4. Under the conditions in Theorem 5 in [Du et al., 2024a] and Proposition 4, with the probability at least $\frac{97}{100} - \frac{7}{3}\delta > 0$

$$\frac{|\{\tilde{\mathbf{x}}_i \in \boldsymbol{X}_{\text{wild}}^{\text{out}} : \tau_i \leq T\}|}{|\boldsymbol{X}_{\text{wild}}^{\text{out}}|} \leq \frac{1 - \min\{1, \Delta(n, m)\}}{1 - T/(\frac{\sqrt{2}}{2t\alpha - 1})^2}, \tag{11}$$

We prove this step: let $Z$ be the uniform random variable with $S_{\text{wild}}^{\text{out}}$ as its support and $Z(i) = \tau_i/(\frac{\sqrt{2}}{2t\alpha - 1})^2$, then by the Markov inequality, we have

$$\frac{|\{\tilde{\mathbf{x}}_i \in \boldsymbol{X}_{\text{wild}}^{\text{out}} : \tau_i < T\}|}{|\boldsymbol{X}_{\text{wild}}^{\text{out}}|} = P(Z(I) < T/(\frac{\sqrt{2}}{2t\alpha - 1})^2) \geq \frac{\Delta(n, m) - T/(\frac{\sqrt{2}}{2t\alpha - 1})^2}{1 - T/(\frac{\sqrt{2}}{2t\alpha - 1})^2}. \tag{12}$$

Step 5. If $\pi \leq \Delta_\xi^\eta/(1 - \epsilon/M')$, then with the probability at least $\frac{97}{100} - \frac{7}{3}\delta > 0$

$$\frac{|\{\tilde{\mathbf{x}}_i \in \boldsymbol{X}_{\text{wild}}^{\text{out}} : \tau_i \leq T\}|}{|\boldsymbol{X}_{\text{wild}}^{\text{out}}|} \leq \frac{\epsilon + (\frac{\sqrt{2}}{2t\alpha - 1})^2 \epsilon'(n, m)}{(\frac{\sqrt{2}}{2t\alpha - 1})^2 - T}, \tag{13}$$

where $\epsilon'(n, m) = \epsilon(n, m)/\pi + \Delta_\xi^\eta \epsilon(m) - \epsilon(n)\epsilon(n, m)$.

Step 6. If we set $\delta = 3/100$, then it is easy to see that

$$\epsilon(m) \leq O(\frac{1}{\pi^2 \sqrt{m}}),$$

$$\epsilon(n, m) \leq O(\beta_1 M \sqrt{\frac{d}{n}}) + O(\beta_1 M \sqrt{\frac{d}{(1 - \pi)m}}),$$

$$\epsilon'(n, m) \leq O(\frac{\beta_1 M}{\pi} \sqrt{\frac{d}{n}}) + O\left((\beta_1 M \sqrt{d} + \sqrt{1 - \pi}\Delta_\xi^\eta/\pi)\sqrt{\frac{1}{\pi^2(1 - \pi)m}}\right).$$

Step 7. By results in Steps 4, 5 and 6, We complete this proof

## C.2 Proof of Theorem 2

The first result is trivial. Hence,we omit it.We mainly focus on the second result in this theorem In this proof, then we set

$$\eta = \sqrt{8\beta_1 R_{\text{in}}^* + 0.99\pi}/(\sqrt{0.98}\sqrt{8\beta_1 R_{\text{in}}^*} + \sqrt{8\beta_1 R_{\text{in}}^* + \pi})$$

Note that it is easy to check that

$$\xi \geq 2.011\sqrt{8\beta_1 R_{\text{in}}^*} + 1.011\sqrt{\pi} \geq \sqrt{8\beta_1 R_{\text{in}}^*} + 1.011\sqrt{8\beta_1 R_{\text{in}}^* + \pi}.$$

Therefore,

$$\eta\xi \geq \frac{1}{\sqrt{0.98}}\sqrt{8\beta_1 R_{\text{in}}^* + 0.99\pi} > \sqrt{8\beta_1 R_{\text{in}}^* + \pi},$$

which implies that $\Delta_\xi^\eta > \pi$ Note that

$$(1 - \eta)\xi \geq \frac{1}{\sqrt{0.98}}\left(\sqrt{0.98}\sqrt{8\beta_1 R_{\text{m}}^*} + \sqrt{8\beta_1 R_{\text{m}}^* + \pi} - \sqrt{8\beta_1 R_{\text{m}}^* + 0.99\pi}\right) > \sqrt{8\beta_1 R_{\text{m}}^*},$$

which implies that $\Delta > 0$ We have completed this proof

# D  Necessary Propositions

## D.1  Boundedness

**Proposition 1.** *If Assumption 1 holds,*

$$\sup_{\mathbf{w}\in\mathcal{W}}\sup_{(\mathbf{x},y)\in \mathbf{X}\times\mathbf{Y}} \|\nabla\ell(h_{\theta_t}(\mathbf{x}),y)\|_2 \leq \beta_1 r_1 + b_1 = \sqrt{M'/2},$$

$$\sup_{\mathbf{w}\in\mathcal{W}}\sup_{(\mathbf{x},y)\in \mathbf{X}\times\mathbf{Y}} \ell(h_{\theta_t}(\mathbf{x}),y) \leq \beta_1 r_1^2 + b_1 r_1 + B_1 = M,$$

*Proof. One can prove this by Mean Value Theorem of Integrals easily.*

**Proposition 2.** *If Assumption 1 holds, for any $\mathbf{w}\in\mathcal{W}$,*

$$\|\nabla\ell(h_{\theta_t}(\mathbf{x}),y)\|_2^2 \leq 2\beta_1\ell(h_{\theta_t}(\mathbf{x}),y).$$

*Proof. The details of the self-bounding property can be found in Appendix B of Lei Ying*

**Proposition 3.** *If Assumption 1 holds, for any labeled data $\mathbf{X}$ and distribution $\mathbb{P}$.*

$$\|\nabla R_{\mathbf{X}}(h_{\theta_t})\|_2^2 \leq 2\beta_1 R_{\mathbf{X}}(h_{\theta_t}), \quad \forall \mathbf{w}\in\mathcal{W}, \tag{14}$$

$$\|\nabla R_{\mathbb{P}}(h_{\theta_t})\|_2^2 \leq 2\beta_1 R_{\mathbb{P}}(h_{\theta_t}), \quad \forall \mathbf{w}\in\mathcal{W}. \tag{15}$$

*Proof. Jensen's inequality implies that $R_S(h_{\theta_t})$ and $R_{\mathbb{P}}(\mathbf{f}_{\mathbf{w}})$ are $\beta_1$ -smooth. Then Proposition 2 implies the results.*

**Proposition 4.** *If Assumption 1 holds, for any $\mathbf{w}_t\in\mathcal{W}$,*

$$\| \Delta h_{\theta_t}(\mathbf{x}) \|_2 \leq (\sqrt{M'/2}+1)/(2t) = \widehat{M_t}$$

*Proof. It is trivial that*

$$\| \mathbf{x}^\top\nabla\ell(h_{\theta_t}(\mathbf{x}),y) \| \leq \| \nabla\ell(h_{\theta_t}(\mathbf{x}),y) \| \leq \beta_1 r_1 + b_1 = \sqrt{M'/2}$$

*Then*

$$\| \mathbf{x}^\top\nabla\ell(h_{\theta_t}(\mathbf{x}),y) \| = \| 2(\mathbf{x}\mathbf{W}^\top - y) \| \geq 2 \| \sum_t \Delta h_{\theta_t}(\mathbf{x}) - y \| \geq 2 \| t\Delta h_{\theta_t}(\mathbf{x}) - y \| \geq 2t \| \Delta h_{\theta_t}(\mathbf{x}) \| - 1$$

*It is straightforward to verify that:*

$$\|\Delta h_{\theta_t}(\mathbf{x})\|_2 \leq \frac{\sqrt{M'/2}+1}{2t} \leq \alpha\sqrt{M'/2} = \widehat{M_t}.$$

*Here, $\alpha$ is the learning rate. From the inequality above, we establish a relationship between $\sqrt{M'/2}$, $\alpha$, and $t$ as follows:*

$$M' \geq (\frac{\sqrt{2}}{2t\alpha - 1})^2.$$

# E  Experiment Details

We supplement experiment details for reproducibility. Our implementation is based on Ubuntu 20.04, Cuda 12.1, Pytorch 2.1.2, and Pytorch Geometric 2.6.1. All the experiments run with an NVIDIA 3090 with 24GB memory.

## E.1  Hyperparameter

As shown in Table 10.

Table 10: Hyper-parameters for training.

| Dataset | Squirrel | WikiCS | YelpChi | Amazon | Reddit | Cora | Citeseer | Pubmed | Chameleon |
|---|---|---|---|---|---|---|---|---|---|
| Learning rate ($\alpha$) | 0.005 | 0.01 | 0.005 | 0.005 | 0.01 | 0.005 | 0.005 | 0.01 | 0.005 |
| $h_\theta$ layers | 1 | 1 | 1 | 1 | 1 | 1 | 1 | 1 | 1 |
| $g_\theta(\cdot)$ layers | 2 | 2 | 2 | 2 | 2 | 2 | 2 | 2 | 2 |
| Hidden states | 16 | 16 | 16 | 16 | 16 | 16 | 16 | 16 | 16 |
| Dropout rate | 0.1 | 0.1 | 0.1 | 0.1 | 0.1 | 0.1 | 0.1 | 0.1 | 0.1 |
| n | 2 | 1 | 2 | 2 | 1 | 10 | 10 | 5 | 2 |
| $\lambda$ | 0.5 | 0.5 | 0.5 | 0.5 | 0.5 | 0.5 | 0.5 | 0.5 | 0.5 |

## E.2 Metric

Following prior research on OOD node detection, we evaluate the detection performance using three widely recognized, threshold-independent metrics: AUROC (↑), AUPR (↑) and FPR95(↓). (1) **AUROC** measures the area under the receiver operating characteristic curve, capturing the trade-off between the true positive rate and the false positive rate across different threshold values. (2) **AUPR** calculates the area under the precision-recall curve, representing the balance between the precision rate and recall rate for OOD nodes across varying thresholds. (3) **FPR95** is defined as the probability that an OOD sample is misclassified as an ID node when the true positive rate is set at 95%.

## E.3 Dataset Description

To thoroughly evaluate the effectiveness of RSL, we perform experiments on real-world node-level and graph-level OOD detection datasets:

- Node-level Datasets:
    - **Squirrel** [Rozemberczki et al., 2021]: A Wikipedia network where nodes correspond to English Wikipedia articles, and edges represent mutual hyperlinks. Nodes are categorized into five classes following Geom-GCN [Pei et al., 2020] annotations, with the network exhibiting a high level of heterophily.
    - **WikiCS** [Mernyei and Cangea, 2020]: This dataset consists of nodes representing articles in the Computer Science domain. Edges are based on hyperlinks, and nodes are classified into 10 categories, each corresponding to a unique sub-field of Computer Science.
    - **YelpChi** [Rayana and Akoglu, 2015]: Derived from Yelp, this dataset includes hotel and restaurant reviews. Legitimate reviews are labeled as ID nodes, while spam reviews are considered OOD nodes.
    - **Amazon** [McAuley and Leskovec, 2013]: Contains reviews from the Musical Instrument category on Amazon.com. ID nodes represent benign users, while OOD nodes correspond to fraudulent users.
    - **Reddit** [Kumar et al., 2019]: A dataset comprising user posts collected from various subreddits over a month. Normal users are treated as ID nodes, while banned users are labeled as OOD nodes.
    - **Chameleon** Rozemberczki et al. [2021] is a Wikipedia network with 5 classes, where nodes represent web pages and edges represent hyperlinks between them. Node features represent several informative nouns in the Wikipedia pages, and the task is to predict the average daily traffic of the web page Fey and Lenssen [2019].
    - **Cora** [Kipf and Welling, 2016a] is a citation graph with 2,708 nodes, 5,429 edges, 1,433 features, and 7 classes, widely used for node classification and link prediction. Under the Label Leave-out setting, 3 classes are treated as ID and 4 as OOD.
    - **Citeseer** [Kipf and Welling, 2016a] contains 3,327 nodes, 4,732 edges, 3,703 features, and 6 classes. We apply the same OOD generation strategies as above, designating 3 classes as ID and 3 as OOD under the Label Leave-out setting.
    - **PubMed** [Kipf and Welling, 2016a], a biomedical citation graph, includes 19,717 nodes, 44,338 edges, 500 features, and 3 classes. We follow the same OOD generation and semi-supervised training procedure, using 2 classes as ID and 1 as OOD under the Label Leave-out setting.

For the Squirrel, WikiCS, YelpChi, Amazon, and Reddit datasets, we follow the same data preprocessing steps as Energy*Def* [Gong and Sun, 2024]. Both Squirrel and WikiCS datasets are

loaded using the DGL [Wang et al., 2019] package. For Squirrel, class {1} is selected as the OOD class, while {0, 2, 3, 4} are designated as ID classes. In the case of WikiCS, {4, 5} are chosen as OOD classes, with the remaining eight classes treated as ID. The YelpChi and Amazon datasets are processed based on the methodology described in [Dou et al., 2020], and the Reddit dataset is prepared using the PyGod [Liu et al., 2022] package. For the Cora and Chameleon datasets, we follow the data processing procedure used in GRASP [Ma et al., 2024].

- Graph-level Datasets:
    - **ENZYMES** Morris et al. [2020] is a graph dataset constructed based on the structural properties of protein molecules. It contains a total of 600 graphs, each representing one protein sample, across six different classes. The dataset includes 19,580 nodes and 174,564 edges, with each node having a feature vector of dimension 3.
    - **PROTEINS** Morris et al. [2020] is a dataset of proteins that are classified as enzymes or non-enzymes. The dataset includes 1,113 graphs.
    - **ClinTox** [Wu et al., 2018] compares drugs approved by the FDA and drugs that have failed clinical trials for toxicity reasons. The dataset includes two classification tasks for 1491 drug compounds with known chemical structures: (1) clinical trial toxicity (or absence of toxicity) and (2) FDA approval status.
    - **LIPO** [Wu et al., 2018] is a dataset included in MoleculeNet [Wu et al., 2018]. It measures the experimental results of octanol/water distribution coefficient(logD at pH 7.4).

We follow the data processing procedure used in GOOD-D [Liu et al., 2023] that 90% of ID samples are used for training, and 10% of ID samples and the same number of OOD samples are integrated together for testing.

## E.4 Baseline Description

- Node-level Baselines:
    - **LOF-KNN** [Breunig et al., 2000] calculates the OOD scores of node attributes by assessing the deviation in local density relative to the k-nearest node attributes.
    - **MLPAE** uses an MLP-based autoencoder, where the reconstruction error of node attributes is used as the OOD score. It is trained by minimizing the reconstruction error on ID training nodes.
    - **GCNAE** [Kipf and Welling, 2016b] swaps the MLP backbone for a GCN in the autoencoder. The OOD score is determined in the same way as MLPAE, following the same training process.
    - **GAAN** [Chen et al., 2020] is a generative adversarial network for attributes that evaluates sample reconstruction error and the confidence of recognizing real samples to predict OOD nodes.
    - **DOMINANT** [Ding et al., 2019] combines a structure reconstruction decoder and an attribute reconstruction decoder. The total reconstruction error for each node consists of the errors from both decoders.
    - **ANOMALOUS** [Peng et al., 2018] is an anomaly detection method that utilizes CUR decomposition and residual analysis for identifying OOD nodes.
    - **SL-GAD** [Zheng et al., 2021] derives OOD scores for nodes by considering two aspects: reconstruction error and contrastive scores.
    - **GOAD** [Bergman and Hoshen, 2020] enhances training data by transforming it into independent spaces and trains a classifier to align the augmented data with the corresponding transformations. OOD scores are then calculated based on the distances between OOD inputs and the centers of the transformation spaces. For graph-structured data, we use the same GNN backbone as EnergyDef-h.
    - **NeuTral AD** [Qiu et al., 2021] uses learnable transformations to embed data into a semantic space. The OOD score is determined by a contrastive loss applied to the transformed data.
    - **MSP** Hendrycks and Gimpel [2016]: Uses the maximum softmax probability as the OOD score. The method is simple but has limited performance on models with high confidence.
    - **ODIN** Liang et al. [2017]: Improves OOD detection by temperature scaling and input perturbation, but is sensitive to hyperparameters.
    - **Mahalanobis** Lee et al. [2018]: Calculates the feature distance between a sample and ID data based on Mahalanobis distance, suitable for scenarios assuming a Gaussian distribution.

- **OE** Hendrycks et al. [2018]: Optimizes using additional OOD data during training, relying on the availability of OOD data.
- **Energy & Energy FineTune** Liu et al. [2020]: Uses an energy function instead of softmax probabilities for OOD scoring, and can improve detection performance by fine-tuning with OOD data.
- **GKDE** [Zhao et al., 2020] predicts Dirichlet distributions for nodes and derives uncertainty as OOD scores by aggregating information from multiple sources.
- **GPN** Stadler et al. [2021]: Based on Bayesian posterior inference, performs OOD detection through uncertainty estimation. It is suitable for graph data but sensitive to hyperparameters.
- **OODGAT** [Song and Wang, 2022] is an entropy-based OOD detector that assumes node category labels are available. It uses a Graph Attention Network as the backbone and determines OOD nodes based on category distribution outcomes.
- **GNNSafe** [Wu et al., 2023] calculates OOD scores by applying the LogSumExp function over the output logits of a GNN classifier, which is trained with multi-category labels. The rationale for the OOD score is the similarity between the Softmax function and the Boltzmann distribution.
- **NodeSafe** [Yang et al., 2025] reduces the occurrence of extreme energy values by enforcing consistency in the logit norms, thereby decreasing the variance within both the ID and OOD energy distributions, which enhances the performance of OOD node detection.
- **GRASP** [Ma et al., 2024] enhances OOD node detection performance by amplifying the graph's homophily through rewiring, thereby improving the effect of score propagation.
- **SSD** [Sehwag et al., 2021] is an outlier detector that leverages self-supervised representation learning and Mahalanobis distance-based detection on unlabeled ID data. We use twice dropout to generate positive pairs for contrastive learning like SimCSE [Gao et al., 2021].
- **Energy*Def*** Gong and Sun [2024] uses Langevin dynamics to generate synthetic OOD nodes for training the OOD node classifier.
- Graph-level Baselines:
  - **Graph kernel + detector**: This category of methods involves two main steps: first, graph kernel techniques are employed to transform graphs into vector-based features Vishwanathan et al. [2010]; next, out-of-distribution (OOD) detection algorithms are applied to these feature vectors. Specifically, we adopt the Weisfeiler-Lehman (WL) kernel Shervashidze et al. [2011] and the propagation kernel (PK) Neumann et al. [2016] for representation, combined with anomaly detectors such as local outlier factor (LOF) Breunig et al. [2000], one-class SVM (OCSVM) Manevitz and Yousef [2001], and isolation forest (iF) Liu et al. [2008].
  - **GCL + detector**: These approaches leverage recent advances in Graph Contrastive Learning (GCL) to derive graph-level embeddings, which are then assessed by OOD detection methods. We employ two representative GCL techniques—InfoGraph Sun et al. [2020] and GraphCL You et al. [2020]—to generate node or graph representations. For detecting OOD instances, we consider both the isolation forest (iF) Liu et al. [2008] and a Mahalanobis distance-based (MD) detector, which has demonstrated strong performance in identifying OOD data Sehwag et al. [2021], Zhou et al. [2021]. GOOD-D Liu et al. [2023] can capture the latent ID patterns and accurately detect OOD graphs based on the semantic inconsistency in different granularities by performing hierarchical contrastive learning on the augmented graphs.
  - **End-to-end**: We also evaluate our model against end-to-end graph anomaly detection baselines. One such approach is OCGIN Zhao and Akoglu [2023], which utilizes a GIN encoder trained with a support vector data description (SVDD) loss. Another is GLocalKD Ma et al. [2022], which detects anomalous samples through a knowledge distillation framework.

### E.5 Implementation Details

We adopt the same dataset settings as Energy*Def* [Gong and Sun, 2024], and we use GCN [Kipf and Welling, 2016a] as the encoder. *It is worth noting that, under this dataset setup, the features of unknown nodes are accessible. Therefore, using the features of unknown nodes during the training phase to filter reliable OOD nodes is a legitimate strategy.* Specifically, for the Squirrel and WikiCS datasets, we randomly select one and two classes as OOD classes, respectively. In the case of fraud detection datasets, we categorize a large number of legitimate entities as ID nodes and fraudsters as

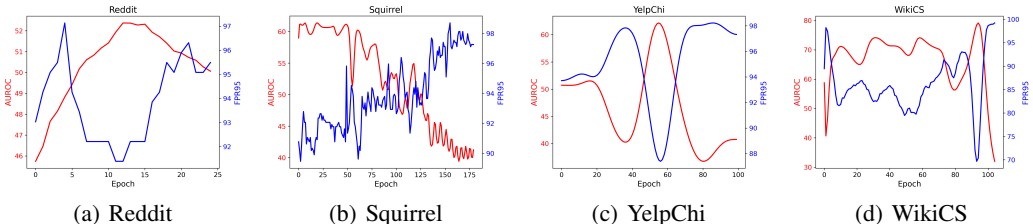

| (a) Reddit | (b) Squirrel | (c) YelpChi | (d) WikiCS |

Figure 3: The performance of using resonance-based score $\tau$ to detect OOD nodes varies with training progress. The higher the AUROC, the better, and the lower the FPR95, the better.

OOD nodes. We allocate 40% of the ID class nodes for training, with the remaining nodes split into a 1:2 ratio for validation and testing, ensuring stratified random sampling based on ID/OOD labels.

We report the average value of five runs for each dataset. The hyper-parameters are shown in Table 10. The anomaly detection baselines are trained entirely based on graph structures and node attributes without requiring ID annotations. We adapt these models to the specifications of our OOD node detection tasks by minimizing the corresponding loss items solely on the ID nodes, where applicable.

## F   More Experiments

### F.1   The Feature Resonance Phenomenon Induced by Different Target Vectors

We explore the phenomenon of feature resonance using different target vectors. Experiments are conducted on two datasets with real $N$-category labels, Squirrel and WikiCS ($N$ represents the number of categories). First, based on the neural collapse theory [Papyan et al., 2020, Zhou et al., 2022], we preset $N$ target vectors, each representing a category. These $N$ target vectors form an equiangular tight frame, maximizing the separation between them. The definition of the simplex equiangular tight frame is introduced as follows:

**Definition 6.** *Simplex ETF. [Xiao et al., 2024] A simplex equiangular tight frame (ETF) refers to a collection of K equal-length and maximally-equiangular P-dimensional embedding vectors* $\mathbf{E} = [e_1, \cdots, e_K] \in \mathbb{R}^{P \times K}$ *which satisfies:*

$$\mathbf{E} = \sqrt{\frac{K}{K-1}} \mathbf{U} \big( \mathbf{I}_K - \frac{1}{K} \mathbf{1}_K \mathbf{1}_K^\top \big) \tag{16}$$

*where* $\mathbf{I}_K$ *is the identity matrix,* $\mathbf{1}_K$ *is an all-ones vector, and* $\mathbf{U} \in \mathbb{R}^{P \times K}(P \geq K)$ *allows a rotation.*

All vectors in a simplex ETF $\mathbf{E}$ have an equal $l_2$ norm and the same pair-wise maximal equiangular angle $-\frac{1}{K-1}$,

$$e_{k_1}^\top e_{k_2} = \frac{K}{K-1} \delta_{k_1,k_2} - \frac{1}{K-1}, \forall k_1, k_2 \in [1, K] \tag{17}$$

where $\delta_{k_1,k_2} = 1$ when $k_1 = k_2$ and 0 otherwise.

We use MSE loss to pull the representations of known ID nodes toward their corresponding target vectors based on their labels, as follows:

$$\ell(h_{\theta_t}(\boldsymbol{X}_{\text{known}}), e) = \mathbb{E}(\| \mathbf{E}_{\text{known}} - (\boldsymbol{X}_{\text{known}} \mathbf{W}^\top) \|_2^2) \tag{18}$$

where $\mathbf{E}_{\text{known}}$ denotes the target vector matrix corresponding to the known ID nodes.

The trajectory trends and lengths of unknown ID nodes differ significantly from those of OOD nodes, with the former showing more distinct trends and longer trajectories. We refer to this as the feature resonance phenomenon and leverage it to filter OOD nodes. As shown in Table 6, under the "True multi-label" row, the experimental results demonstrate that this method is effective and performs well. Interestingly, even with random labels for known ID nodes or aligning all known ID representations to a fixed target vector, unknown ID nodes consistently exhibit longer trajectories than unknown OOD nodes, as shown in Table 6.

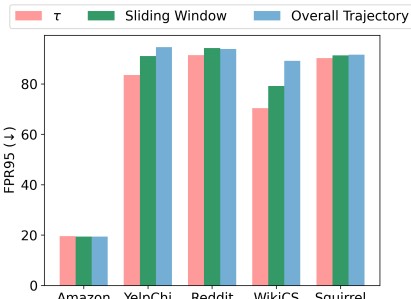

Figure 4: Performance of detecting OOD nodes with different metrics. $\tau$ represents the resonance-based score, the "Overall Trajectory" represents the total cumulative length of the training trajectory $\hat{F}(\tilde{\mathbf{x}}_i) = \sum_t \tau_i$, and the "Sliding Window" refers to the cumulative $\tau$ within a window of width 10: $\hat{F}_{10}(\tilde{\mathbf{x}}_i) = \sum_{t-10}^{t} \tau_i$.

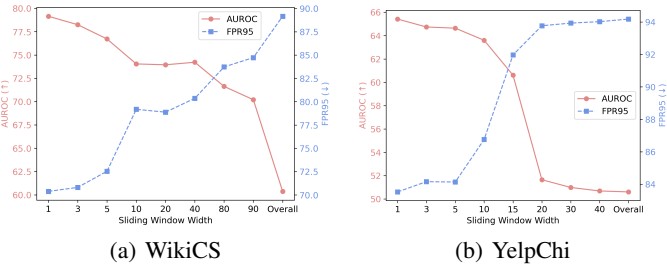

(a) WikiCS        (b) YelpChi

Figure 5: The impact of different sliding window widths on the performance of detecting OOD nodes. When the width is 1, it corresponds to the resonance-based score $\tau$.

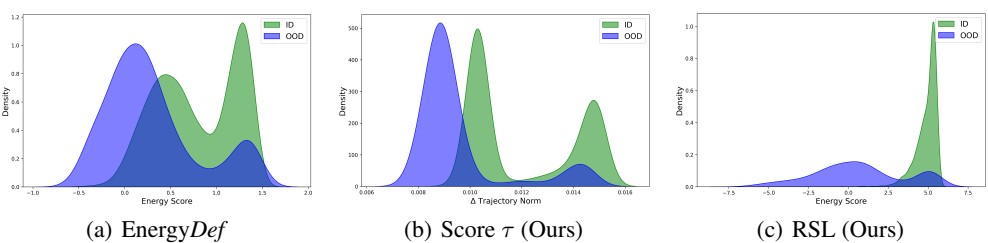

(a) Energy*Def*      (b) Score $\tau$ (Ours)      (c) RSL (Ours)

Figure 6: The score distribution of ID nodes and OOD nodes on *Amazon*.

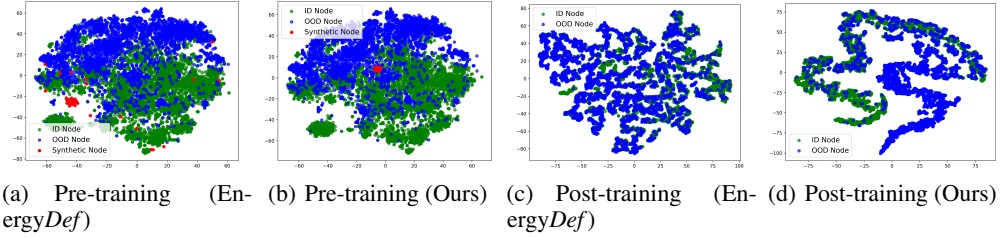

(a) Pre-training (Energy*Def*)   (b) Pre-training (Ours)   (c) Post-training (Energy*Def*)   (d) Post-training (Ours)

Figure 7: T-SNE visualization of node embeddings on the dataset *WikiCS*. (a) Synthetic nodes (red) generated by Energy*Def* fail to accurately represent the actual features of OOD nodes (blue), whereas ours can, as shown in (b). (c) Representations of ID (green) and OOD (blue) nodes trained with synthetic nodes generated by Energy*Def* are poorly separated, whereas ours can, as shown in (d).

The experiments above indicate that the feature resonance phenomenon is *label-independent* and results from the intrinsic relationships between ID node representations. Therefore, this is highly suitable for category-free OOD detection scenarios without multi-category labels.

Table 11: Time cost (s).

| Method \ Dataset | Squirrel | WikiCS | YelpChi | Amazon | Reddit |
|---|---|---|---|---|---|
| EnergyDef | 10.94 | 27.11 | 76.51 | 33.81 | 26.44 |
| RSL w/o classifier | 5.25 | 4.03 | 5.41 | 5.75 | 3.71 |
| RSL | 11.54 | 17.53 | 74.83 | 36.33 | 38.23 |

Table 12: Comparison on WikiCS and Amazon datasets using different GNN encoders.

| GNN Encoder | Method | WikiCS | | | Amazon | | |
|---|---|---|---|---|---|---|---|
| | | AUROC↑ | AUPR↑ | FPR@95↓ | AUROC↑ | AUPR↑ | FPR@95↓ |
| GCN | EnergyDef | 70.22 | 60.10 | 83.17 | 86.57 | 74.50 | 32.43 |
| GCN | RSL | **84.01** | **81.14** | **49.23** | **90.03** | **83.91** | **19.60** |
| GAT | EnergyDef | 74.22 | 64.15 | 79.80 | 88.20 | 78.40 | 27.88 |
| GAT | RSL | **88.01** | **86.37** | **41.02** | **88.28** | **82.90** | **20.48** |
| GIN | EnergyDef | 72.18 | 62.35 | 80.56 | 85.98 | 75.10 | 31.47 |
| GIN | RSL | **83.74** | **82.58** | **43.50** | **91.58** | **84.39** | **19.74** |

Table 13: Performance comparison across different methods on graph-level OOD detection (AUROC).

| Model | ID: ENZYMES / OOD: PROTEIN | ID: ClinTox / OOD: Lipo |
|---|---|---|
| PK-LOF | 50.47±2.87 | 50.00±2.17 |
| PK-OCSVM | 50.46±2.78 | 50.06±2.19 |
| PK-iF | 51.67±2.69 | 50.81±1.10 |
| WL-LOF | 52.66±2.47 | 51.29±3.40 |
| WL-OCSVM | 51.77±2.21 | 50.77±3.69 |
| WL-iF | 51.17±2.01 | 50.41±2.17 |
| InfoGraph-iF | 60.00±1.83 | 48.51±1.87 |
| InfoGraph-MD | 55.25±3.51 | 48.12±5.72 |
| GraphCL-iF | 61.33±2.27 | 47.84±0.92 |
| GraphCL-MD | 52.87±6.11 | 51.58±3.64 |
| OCGIN | 57.65±2.96 | 49.13±4.13 |
| GLocalKD | 57.18±2.03 | 55.71±3.81 |
| GOOD-D$_{simp}$ | 61.89±2.51 | 66.13±2.98 |
| GOOD-D | 61.84±1.94 | 69.18±3.61 |
| **RSL (ours)** | **62.53**±1.89 | **72.03**±2.87 |

## F.2 Variation of Microscopic Feature Resonance During Training

We also observe the variation of the microscopic feature resonance phenomenon during the training process on other datasets, as shown in Figure 3. We find that the changes on Reddit, YelpChi, and WikiCS are generally consistent with Amazon, with the most significant feature resonance occurring in the middle of the training process. However, for Squirrel, the feature resonance phenomenon reaches its most pronounced level early in the training. We believe this is due to the relatively rich features in Squirrel, which allow the model to quickly identify the optimal optimization path for ID samples in the early stage of training.

## F.3 Effectiveness of Different Scoring Strategies Based on Feature Resonance

We evaluate the effectiveness of three score design strategies based on feature resonance: the resonance-based score $\tau$, the global trajectory norm, and the sliding window accumulation (width 10). As shown in Figure 4, $\tau$ outperforms the other two scores on most datasets. The sliding window approach performs better than the global trajectory norm. The experimental results in Figure 5 show that as the sliding window width increases, the detection performance for OOD nodes gradually decreases, which indicates that finer-grained information improves OOD node detection, so we select $\tau$ as the primary score for filtering OOD nodes in our method.

## F.4 Time Efficiency

We compare the time consumption of our method, RSL, with the current SOTA method, EnergyDef. The experimental results are shown in Table 11. The experiments show that the overall time efficiency of RSL is comparable to that of EnergyDef, with similar time consumption across different datasets. However, it is worth noting that when we use the resonance-based score $\tau$ alone for OOD node

Table 14: **Results on CIFAR-10.** Comparison with competitive OOD detection methods. ↑ indicates larger values are better and vice versa.

| Method | SVHN | | LSUN | | iSUN | | Texture | | Average | |
|---|---|---|---|---|---|---|---|---|---|---|
| | FPR↓ | AUROC↑ | FPR↓ | AUROC↑ | FPR↓ | AUROC↑ | FPR↓ | AUROC↑ | FPR↓ | AUROC↑ |
| **Without Contrastive Learning** | | | | | | | | | | |
| MSP | 59.66 | 91.25 | 45.21 | 93.80 | 54.57 | 92.12 | 66.45 | 88.50 | 56.47 | 91.42 |
| ODIN | 53.78 | 91.30 | 10.93 | 97.93 | 28.44 | 95.51 | 55.59 | 89.47 | 37.19 | 93.55 |
| Energy | 54.41 | 91.22 | 10.19 | 98.05 | 27.52 | 95.59 | 55.23 | 89.37 | 36.83 | 93.56 |
| GODIN | 18.72 | 96.10 | 11.52 | 97.12 | 30.02 | 94.02 | 33.58 | 92.20 | 23.46 | 94.86 |
| Mahalanobis | 9.24 | 97.80 | 67.73 | 73.61 | 6.02 | 98.63 | 23.21 | 92.91 | 26.55 | 90.74 |
| KNN | 27.97 | 95.48 | 18.50 | 96.84 | 24.68 | 95.52 | 26.74 | 94.96 | 24.47 | 95.70 |
| FR $\tau$ (ours) | 23.50 | 94.85 | 11.48 | 97.80 | 20.93 | 95.67 | 29.22 | 95.28 | 21.28 | 95.90 |
| **With Contrastive Learning** | | | | | | | | | | |
| CSI | 37.38 | 94.69 | 5.88 | 98.86 | 10.36 | 98.01 | 28.85 | 94.87 | 20.62 | 96.61 |
| SSD+ | 1.51 | 99.68 | 6.09 | 98.48 | 33.60 | 95.16 | 12.98 | 97.70 | 13.55 | 97.76 |
| KNN+ | 2.42 | 99.52 | 1.78 | 99.48 | 20.06 | 96.74 | 8.09 | 98.56 | 8.09 | 98.56 |
| FR $\tau$ (ours) | 3.27 | 99.34 | 0.44 | 99.84 | 9.24 | 98.23 | 14.57 | 97.28 | **6.88** | **98.67** |

Table 15: **Evaluation on hard OOD detection tasks.** Model is trained on CIFAR-10 with SupCon loss.

| Method | LSUN-FIX | | | ImageNet-FIX | | | ImageNet-R | | | Average | | |
|---|---|---|---|---|---|---|---|---|---|---|---|---|
| | AUROC↑ | AUPR↑ | FPR↓ | AUROC↑ | AUPR↑ | FPR↓ | AUROC↑ | AUPR↑ | FPR↓ | AUROC↑ | AUPR↑ | FPR↓ |
| SSD+ | 95.52 | 96.47 | 29.88 | 94.85 | 95.77 | 32.29 | 93.40 | 94.93 | 45.88 | 94.59 | 95.72 | 36.02 |
| KNN+ | 96.51 | 97.20 | 21.54 | 95.71 | 96.37 | 25.93 | 95.08 | 95.95 | 30.20 | 95.77 | 96.51 | 25.89 |
| FR $\tau$ (ours) | 96.41 | 97.10 | 21.80 | 95.13 | 95.66 | 26.76 | 97.33 | 97.74 | 15.27 | **96.29** | **96.83** | **21.28** |

detection, its efficiency improves significantly over Energy*Def*, with an average reduction of **79.81%** in time consumption. This indicates that $\tau$ not only demonstrates significant effectiveness in detecting OOD nodes but also offers high efficiency.

### F.5 RSL Performance with Different GNN Encoders

We conduct experiments on WikiCS and Amazon datasets using GCN [Kipf and Welling, 2016a], GAT [Velickovic et al., 2018], and GIN [Xu et al., 2019] as the encoders, respectively. The results in Table 12 show that our RSL consistently outperforms the state-of-the-art method Energy*Def* across all settings.

### F.6 Graph-Level OOD Detection

Since RSL can be easily extended to independent samples beyond nodes—such as entire graphs—we aim to evaluate its performance on graph-level OOD detection tasks. The results in Table 13 show that RSL maintains strong performance in this setting. On the ENZYMES, PROTEINS, ClinTox, and Lipo datasets, RSL outperforms the previous strong baseline, GOOD-D. This highlights the superior scalability of RSL, demonstrating that it is not limited to node-level OOD detection.

### F.7 The Generality of Feature Resonance

While our method is developed in the graph context, its core idea stems from representation dynamics rather than graph-specific structural properties. Nevertheless, graph structure—particularly homophily—can influence feature evolution and thus affect the resonance patterns. As shown in Table 5 of the main paper, higher graph homophily correlates with more pronounced node feature resonance and improved OOD node detection performance. This is because greater homophily generally results in higher-quality node representations. Therefore, the feature resonance phenomenon itself is not solely dependent on the graph structure. To demonstrate its generality, we apply our method to standard image datasets following the setup in Sun et al. [2022] strictly—using image representations extracted from ResNet-18 models trained on CIFAR-10 with either cross-entropy loss ("without contrastive learning") or supervised contrastive learning ("with contrastive learning"). We induce resonance by aligning known ID features to a target vector consisting of the mean of all known ID

samples and measure step-wise changes in unknown samples. As shown in Table 14, our method remains effective on images. Models with stronger initialization (via contrastive learning) exhibit more pronounced resonance, consistent with Table 4 of the main paper. Further, Table 15 shows strong performance on a more challenging image OOD benchmark. We also evaluate our method on graph-level OOD detection (Table 13), where representations are independent like in images, and observe similarly strong results—supporting the universality of the feature resonance phenomenon.

### F.8 Score Distribution Visualization

We visualize the score distributions of ID and OOD nodes on the Amazon dataset obtained using different methods, as shown in Figure 6. When using the resonance-based score (Figure 6 (b)), the majority of unknown ID nodes show more significant representation changes compared to unknown OOD nodes. This separation of OOD nodes already exceeds Energy*Def* (Figure 6 (a)). After training with synthetic OOD nodes (Figure 6 (c)), the separation between the energy scores of ID and OOD nodes still improves compared to Energy*Def*, which demonstrates the effectiveness of RSL.

### F.9 Node Representation Visualization

Energy*Def* generates auxiliary synthetic OOD nodes via SGLD to train an OOD classifier for category-free OOD node detection. However, we find that the synthetic OOD nodes from Energy*Def* do not accurately capture the features of actual OOD nodes. As shown in Figure 7(a), most synthetic OOD nodes are separated from actual OOD nodes and even overlap with ID nodes, limiting the classifier's performance. The severe overlap between ID and OOD node representations after training by Energy*Def* (Figure 7(c)) further highlights this issue. In contrast, we use feature resonance to identify reliable OOD nodes and synthesize new ones based on these. As seen in Figure 7(b), our synthetic OOD nodes align more closely with the actual OOD nodes. Training with these nodes results in better separation between ID and OOD node representations, as shown in Figure 7(d).

## G  Discussion

### G.1  A Straightforward Explanation of Feature Resonance

To verify the phenomenon of Feature Resonance, we calculate the change $\Delta h_{\theta_t}(\tilde{\mathbf{x}}_i)$ in the representation $h_{\theta_t}(\tilde{\mathbf{x}}_i)$ of an unlabeled node $i$ from the $t$-th ($t \geq 0$) epoch to the $(t+1)$-th epoch, defined as follows:

$$
\begin{aligned}
&\Delta h_{\theta_t}(\tilde{\mathbf{x}}_i) \\
&= h_{\theta_{t+1}}(\tilde{\mathbf{x}}_i) - h_{\theta_t}(\tilde{\mathbf{x}}_i) \\
&= -\alpha \, \tilde{\mathbf{x}}_i \, \nabla_{\theta_t} \ell(\boldsymbol{X}_{\text{known}}) \\
&= 2\alpha \mathbb{E}(\underbrace{\tilde{\mathbf{x}}_i \boldsymbol{X}_{\text{known}}^{\top}}_{\text{Term 1}} (\underbrace{(\boldsymbol{X}_{\text{known}} \mathbf{W}_t^{\top}) - \mathbf{1}^{\top} e}_{\text{Term 2}}))
\end{aligned}
\tag{19}
$$

where $\alpha$ is the learning rate. Term 1 in the Equation 19 illustrates that when the features of $\tilde{\mathbf{x}}_i$ are consistent with the overall features of the labeled ID nodes $\boldsymbol{X}_{\text{known}}$, the representation of $\tilde{\mathbf{x}}_i$ undergoes a more significant change. Meanwhile, since term 2 in the Equation 19 and $\tilde{\mathbf{x}}_i$ are independent, the choice of the target vector can be arbitrary. It is highly suitable for category-free OOD detection scenarios, requiring no multi-category labels as ground truth.

### G.2  Why Feature Resonance Tends to Occur in the Middle Stages of Training

Empirically, we observe that **feature resonance peaks in the middle of training**. Although it is challenging to fully explain why feature resonance is most prominent in the middle stages of training, we aim to provide some theoretical insights. This aligns with the **Information Bottleneck (IB) theory** Tishby and Zaslavsky [2015], Saxe et al. [2019] and recent feature learning studies Allen-Zhu and Li [2022], Cao et al. [2022], which suggest that *models initially memorize broad information, then gradually compress irrelevant parts while preserving task-relevant features*—reflecting an emerging inductive bias. This compression phase in the middle of training corresponds to a point where irrelevant variation is reduced, allowing feature resonance to become most salient.

According to the IB principle, the representation $T$ is optimized by:

$$\min_{p(T|X)} I(X;T) - I(T;Y), \tag{20}$$

where $I(X;T)$ measures how much input information is retained, and $I(T;Y)$ indicates task relevance.

The training dynamics can be interpreted as follows:

1. **Early training:** $I(X;T) \uparrow$, $I(T;Y)$ is low $\Rightarrow$ large information redundancy, unstable representations, and little or no resonance.
2. **Middle training:** $I(X;T) \downarrow$, $I(T;Y) \uparrow \Rightarrow$ irrelevant information is compressed, task-relevant features are amplified, resulting in strong feature resonance.
3. **Late training:** possible overfitting, $I(X;T) \uparrow$ again, but no further gain in $I(T;Y) \Rightarrow$ representations become more complex, and feature resonance diminishes.

This dynamic explains why feature resonance tends to emerge most clearly during the middle stages of training.

### G.3 Differences from Gradient-Based Methods

It is important to note that our method RSL differs significantly from previous gradient-based methods:

*1) Originating from the Commonality of Representations.* Our method is based on the conjecture that there are inherent commonalities between the representations of the ID sample, which are independent of gradients.

*2) No Pre-trained Multi-category Classifier Required.* Gradient-based methods like GradNorm [Huang et al., 2021] compute the KL divergence between an unknown sample's softmax output from a multi-category classifier and a uniform distribution, using the gradient norm to distinguish OOD samples. OOD samples, with uniform softmax outputs, yield more minor gradient norms, whereas sharper outputs for ID samples produce more significant norms. Similarly, SAL [Du et al., 2024a] uses pseudo-labels from a multi-category classifier for unknown samples, continuing training to compute gradients, and identifies OOD samples via the gradient's principal component projection. These methods require a pre-trained multi-category classifier, making them unsuitable for category-free scenarios without labels, whereas our RSL method avoids this limitation.

*3) No Need to Compute Gradients for Unknown Samples.* As shown in Equation 19, we only need the representations of unknown samples to compute our resonance-based score. This significantly enhances the flexibility of our method, as we can detect OOD samples during any optimization of known ID representations without the need to wait until after the optimization is complete.

### G.4 Limitations and Future Directions

While our method demonstrates strong performance in OOD detection within the graph domain, its applicability and effectiveness in other domains, such as computer vision (CV), natural language processing (NLP), and multimodal data, remain largely unexplored. These domains often come with distinct data structures, noise characteristics, and task-specific challenges, which may affect the dynamics of feature resonance and the general behavior of our approach. Future work could investigate how the core principles of RSL, such as feature resonance and unsupervised separation of ID and OOD distributions, translate to domains where data representations are less structured or more abstract than in graphs.

Although our use of the validation set strictly follows the setting of the latest baseline EnergyDef [Gong and Sun, 2024], where the validation and test sets are constructed by randomly splitting the unknown ID and OOD nodes at a 1:2 ratio, the roles of the validation set differ slightly in our approach. In EnergyDef [Gong and Sun, 2024], the validation set is used solely for selecting the best checkpoint. In our method, however, it serves two purposes: during Stage 1, it is used to determine the optimal threshold $t$ when identifying high-confidence OOD nodes through feature resonance, and during Stage 2, it is used to select the best checkpoint for training the OOD classifier. Nevertheless,

we acknowledge that relying on a validation set introduces certain limitations, and in future work, we aim to develop a feature resonance–based induction method that can operate without the need for a validation set.

Moreover, as a general-purpose algorithm originally designed for unsupervised scenarios, RSL inherently does not rely on label information. In situations where label data is available, especially in high-resource settings, it currently leverages such information only indirectly through node or sample features. However, this represents an opportunity rather than a limitation. A more deliberate integration of label signals could significantly enhance the learning process, especially for improving the discriminability between ID and OOD instances.

One promising direction is to augment RSL with lightweight supervision or semi-supervised techniques. For example, incorporating label propagation methods could help better spread the influence of known ID categories across the feature space, strengthening the boundary between in-distribution and out-of-distribution regions. Other techniques, such as consistency regularization, pseudo-labeling, or contrastive learning guided by label information, may also be explored to bridge the gap between unsupervised robustness and label-aware precision.

### G.5   Broader Impacts

On the positive side, our method can enhance the robustness and reliability of graph-based systems in various applications, such as fraud detection, cybersecurity, and scientific discovery, by identifying anomalous or out-of-distribution nodes without relying on labels. This has the potential to improve safety and trust in real-world systems.

On the other hand, we recognize that misuse of OOD detection—such as for unjustified surveillance or exclusion of minority data—could raise ethical concerns. To mitigate such risks, we emphasize the importance of transparent usage, fairness-aware evaluation, and domain-specific safeguards. While our method is label-free and unsupervised, it is critical to apply it with caution, particularly in sensitive or high-stakes domains.

## H Algorithm Pseudo-code

---

**Algorithm 1** Resonance-based Separate and Learn (RSL) Framework for Category-Free OOD Detection

---

1: **Input:** Known ID nodes $\mathcal{V}_{\mathrm{known}}$, Wild nodes $\mathcal{V}_{\mathrm{wild}}$, Target vector $e$ with random initial, Validation set $\mathcal{V}_{\mathrm{val}}$
2: **Output:** OOD classifier $E_\theta$
3: **Phase 1: Feature Resonance Phenomenon**
4: Initialize model $h_\theta$ with random parameters $\theta$
5: **for** $t = 1$ to $\mathbb{T}$ (training epochs) **do**
6:     Optimize $h_{\theta_t}(\cdot)$ to align $\mathcal{V}_{\mathrm{known}}$ with target $e$:

$$\ell(h_{\theta_t}(\boldsymbol{X}_{\mathrm{known}}), e) = \mathbb{E}(\|\ \mathbf{1}^\top e - (\boldsymbol{X}_{\mathrm{known}}\mathbf{W}^\top)\ \|_2^2)$$

7:     Calculate the representation change of $\tilde{v}_i \in \mathcal{V}_{\mathrm{wild}} : \Delta h_{\theta_t}(\tilde{\mathbf{x}}_i) = h_{\theta_{t+1}}(\tilde{\mathbf{x}}_i) - h_{\theta_t}(\tilde{\mathbf{x}}_i)$
8:     Compute resonance-based score $\tau_i = \|\ \Delta h_{\theta_t}(\tilde{\mathbf{x}}_i)\ \|_2$
9: **end for**
10: Identify the period of feature resonance using the validation set, selecting $t$ where $\tau$ best separates ID and OOD nodes.
11: **Phase 2: Candidate OOD Node Selection**
12: Define candidate OOD set:

$$\mathcal{V}_{\mathrm{cand}} = \{\tilde{v}_i \in \mathcal{V}_{\mathrm{wild}} : \tau_i \leq T\}$$

13: **Phase 3: Synthetic OOD Node Generation**
14: **for** each $\hat{v}_j \in \mathcal{V}_{\mathrm{syn}}$ (synthetic OOD nodes) **do**
15:     Generate $\hat{\mathbf{x}}_j^{(t+1)}$ with random initial using:

$$\hat{\mathbf{x}}_j^{(t+1)} = \lambda\big(\hat{\mathbf{x}}_j^{(t)} - \frac{\alpha}{2}\nabla_{\hat{\mathbf{x}}_j^{(t)}} E_\theta(\hat{v}_j^{(t)}) + \epsilon\big) + (1-\lambda)\mathbb{E}_{\mathbf{x}\sim\boldsymbol{X}_{\mathrm{cand}}}(\mathbf{x} - \hat{\mathbf{x}}_j^{(t)}),, \epsilon \sim \mathcal{N}(0, \zeta)$$

16: **end for**
17: **Phase 4: OOD Classifier Training**
18: Define training set $\mathcal{V}_{\mathrm{train}} = \mathcal{V}_{\mathrm{known}} \cup \mathcal{V}_{\mathrm{cand}} \cup \mathcal{V}_{\mathrm{syn}}$
19: Assign labels $\boldsymbol{Y}_{\mathrm{train}}$ for ID nodes (1) and OOD nodes (0)
20: Train $E_\theta$ using binary cross-entropy loss:

$$\ell_{\mathrm{cls}} = \mathbb{E}_{v\sim\mathcal{V}_{\mathrm{train}}}\big(\mathrm{y}_v\log(\sigma(E_\theta(v))) + (1-\mathrm{y}_v)\log(1 - \sigma(E_\theta(v)))\big)$$

21: **Return:** Trained OOD classifier $E_\theta$

---

