# OpenReview forum: "Harnessing Feature Resonance under Arbitrary Target Alignment for Out-of-Distribution Node Detection"
_NeurIPS.cc/2025/Conference — NeurIPS 2025 poster_

### Official Review · Reviewer_svYR · 2025-06-25

**Clarity:** 3
**Significance:** 3
**Originality:** 3
**Rating:** 4
**Confidence:** 4

**Summary:**

This paper tackles the important challenge of out-of-distribution (OOD) node detection in graphs without relying on label supervision or predefined pretext tasks, which limits existing methods. The authors introduce a novel perspective based on Feature Resonance, observing that OOD nodes exhibit less representation change than unknown in-distribution nodes during training, even without labels. Building on this insight, they propose RSL, a framework that quantifies feature movement as a proxy for resonance and leverages synthetic OOD nodes to train an effective classifier. Theoretical analysis and extensive experiments on thirteen real-world datasets demonstrate the method’s strong separability and state-of-the-art performance.

**Questions:**

Please see the weaknesses.

**Ethical Concerns:**

["NO or VERY MINOR ethics concerns only"]

**Final Justification:**

After reviewing the other reviewers' comments and authors' response, I have decided to maintain my positive rating.

**Quality:**

3

**Strengths And Weaknesses:**

Strengths:
1. The core idea is good—the observation that ID and OOD samples follow different feature evolution trajectories, and leveraging this for OOD detection is both novel and interesting.
2. The paper provides a degree of theoretical analysis to support its methodology.
3. The visualizations are clear and effectively illustrate the proposed concepts.

Weaknesses:
1. The phenomenon presented in Section 2 is demonstrated under a labeled setting, raising concerns about whether it still holds in the label-free scenario that the paper aims to address.
2. The connection between the observed phenomenon and graph-structured data is not sufficiently discussed. eg,. Is this phenomenon universal? Is it related to structural information?
3. The experimental setup lacks clarity, particularly regarding how the detection stage t is determined and how OOD nodes are defined and separated.

---

> ### Author Rebuttal · Authors · 2025-07-30
>
> Thanks very much for your insightful comments and suggestions! Our detailed responses are as follows.
>
> >**W1:** The phenomenon presented in Section 2 is demonstrated under a labeled setting, raising concerns about whether it still holds in the label-free scenario that the paper aims to address.
> >**A:**
>  The labels in Section 2.1 are used purely for explanatory purposes—**to verify and visualize the existence of feature resonance** by comparing ID and OOD groups. However, our actual approach in Section 2.2 does **not rely on any multi-class labels**. Instead, it leverages the alignment of all known ID representations to an arbitrary fixed target vector—meaning a vector chosen independently of any class labels—to induce and observe micro-level feature dynamics such as step-wise representation shifts. We have conducted extensive experiments on real-world benchmarks (Amazon, Squirrel, WikiCS, YelpChi, Reddit), which show **consistent micro-level resonance patterns**, where ID and OOD nodes differ in representation dynamics across training (Figure 2 and Figure 3 of the Appendix).
>
>
>
> >**W2:** The connection between the observed phenomenon and graph-structured data is not sufficiently discussed. eg,. Is this phenomenon universal? Is it related to structural information?
> >**A:**  Insightful question!  While our method is developed in the graph context, **its core idea stems from representation dynamics** rather than graph-specific structural properties. Nevertheless, graph structure—particularly homophily—can influence feature evolution and thus affect the resonance patterns. **As shown in Table 6 of the main paper**, higher graph homophily correlates with more pronounced node feature resonance and improved OOD node detection performance. This is because **greater homophily generally results in higher-quality node representations**. Therefore, the feature resonance phenomenon itself is not solely dependent on the graph structure.
> To demonstrate its generality, we apply our method to standard image datasets **following the setup in [1] strictly**—using image representations extracted from ResNet-18 models trained on CIFAR-10 with either cross-entropy loss ("without contrastive learning") or supervised contrastive learning ("with contrastive learning"). We induce resonance by aligning known ID features to a random target vector and measure step-wise changes in unknown samples. **As shown in Table 1 below**, our method remains effective on images. Models with stronger initialization (via contrastive learning) exhibit more pronounced resonance, **consistent with Table 5 of the main paper**. Further, Table 2 below shows strong performance on a more challenging image OOD benchmark. We also evaluate our method on **graph-level OOD detection (Appendix Table 13)**, where representations are independent like in images, and observe similarly strong results—**supporting the universality of the feature resonance phenomenon**.
>
> **Table 1. Results on CIFAR-10.** Comparison with competitive OOD detection methods.  ↑ indicates larger values are better and vice versa.
> |Method|SVHN|SVHN|LSUN|LSUN|iSUN|iSUN|Texture|Texture|Average|Average|
> |-|-|-|-|-|-|-|-|-|-|-|
> || FPR↓|AUROC↑|FPR↓|AUROC↑|FPR↓|AUROC↑|FPR↓|AUROC↑|FPR↓|AUROC↑|
> |||||**Without Contrastive Learning**|
> |MSP|59.66|91.25|45.21|93.80|54.57|92.12|66.45|88.50|56.47|91.42|
> |ODIN|53.78|91.30|10.93|97.93|28.44|95.51|55.59|89.47|37.19|93.55|
> |Energy|54.41|91.22|10.19|98.05|27.52|95.59|55.23|89.37|36.83|93.56|
> |GODIN|18.72|96.10|11.52|97.12|30.02|94.02|33.58|92.20|23.46|94.86|
> |Mahalanobis|9.24|97.80|67.73|73.61|6.02|98.63|23.21|92.91|26.55|90.74|
> |KNN|27.97|95.48|18.50|96.84|24.68|95.52|26.74|94.96|24.47|95.70|
> |FR $\tau$ (ours)|23.50|94.85|11.48|97.80|20.93|95.67|29.22|95.28|21.28|95.90|
> |||||**With Contrastive Learning** |
> |CSI|37.38|94.69|5.88|98.86|10.36|98.01|28.85|94.87|20.62|96.61|94.38|
> |SSD+|1.51|99.68|6.09|98.48|33.60|95.16|12.98|97.70|13.55|97.76|**95.07**|
> |KNN+|2.42|99.52|1.78|99.48|20.06|96.74|8.09|98.56|8.09|98.56|**95.07**|
> |FR $\tau$ (ours)|3.27|99.34|0.44|99.84|9.24|98.23|14.57|97.28|**6.88**|**98.67**|**95.07**|
>
> **Table 2.** Evaluation on hard OOD detection tasks. The model is trained on CIFAR-10 with SupCon loss.
> |Method|LSUN-FIX |LSUN-FIX|LSUN-FIX|ImageNet-FIX|ImageNet-FIX|ImageNet-FIX|ImageNet-R|ImageNet-R|ImageNet-R|Average|Average|Average|
> |-|-|-|-|-|-|-|-|-|-|-|-|-|
> ||AUROC↑|AUPR↑|FPR↓|AUROC↑|AUPR↑|FPR↓|AUROC↑|AUPR↑|FPR↓|AUROC↑|AUPR↑|FPR↓|
> |SSD+|95.52|96.47|29.88|94.85|95.77|32.29|93.40|94.93|45.88|94.59|95.72|36.02|
> |KNN+|96.51|97.20|21.54|95.71|96.37|25.93|95.08|95.95|30.20|95.77|96.51|25.89|
> |FR $\tau$ (ours)|96.41|97.10|21.80|95.13|95.66|26.76|97.33|97.74|15.27|**96.29**|**96.83**|**21.28**|
>
>
> >**W3:** The experimental setup lacks clarity, particularly regarding how the detection stage $t$ is determined and how OOD nodes are defined and separated.
> >**A:**  Thank you for your thoughtful comments! We clarify the experimental setup as follows: Our experimental setup strictly follows the latest baseline EnergyDef [2] to ensure consistency. **Detailed configurations are provided in Appendix E.** The detection stage $t$ refers to the epoch at which micro-level feature resonance is most prominent. **We do not manually set this value**—instead, $t$ is automatically selected based on performance on a small validation set. **Our use of the validation set strictly follows the setup of the latest baseline EnergyDef [2]**, where the validation and test sets are constructed by randomly splitting the unknown ID and OOD nodes in a 1:2 ratio. In EnergyDef [2], the validation set is used to select the best checkpoint. In our method, the validation set serves **two roles**: during Stage 1, it is used to select the optimal $t$ when mining high-confidence OOD nodes via feature resonance; during Stage 2, it is used to select the best checkpoint for training the OOD classifier. **Importantly, this strategy of selecting the best model based on validation performance is also widely adopted in prior OOD detection works [2-7].**
> As for how OOD nodes are defined and separated, during stage one, we use the resonance-based score to rank all unlabeled nodes and select the high-confidence OOD samples. These are used in stage two for training the OOD classifier and as anchors to synthesize additional OOD representations. We will add clarifications in the revised version to make these points more explicit.
>
>
> [1] Sun Y, et al. "Out-of-distribution detection with deep nearest neighbors." ICML'22
> [2] Gong Z, et al. "An energy-centric framework for category-free out-of-distribution node detection in graphs." KDD'24.
> [3] Zheng Y, et al. "Generative and contrastive self-supervised learning for graph anomaly detection." TKDE'21.
> [4] Bergman L, et al. "Classification-based anomaly detection for general data." ICLR'20.
> [5] Qiu C, et al. "Neural transformation learning for deep anomaly detection beyond images." ICML'21
> [6] Zhao X, et al. "Uncertainty aware semi-supervised learning on graph data." NIPS'20
> [7] Song Y, et al. "Learning on graphs with out-of-distribution nodes." KDD'22

---

> > ### Comment · Reviewer_svYR · 2025-08-05
> >
> > Thank you for your detailed response. I appreciate the OOD detection experiments on image datasets and their findings on the relationship between homophily and detection performance (theoretical support would be better, but it is beyond the scope of this paper). I have no further questions and will maintain my positive rating.

---

> > > ### Author Response · Authors · 2025-08-05
> > >
> > > Thank you so much for your reply! We appreciate your time for providing insightful comments, which definitely help us a lot for improving our work.

---

### Official Review · Reviewer_J96B · 2025-07-02

**Clarity:** 2
**Significance:** 3
**Originality:** 3
**Rating:** 4
**Confidence:** 5

**Summary:**

Existing graph Out-of-distribution (OOD) node detection methods face a critical limitation by heavily depending on labeled data and well-defined pretext classification tasks. To overcome this, the paper introduces ​​RSL, a novel framework exploiting a newly discovered ​​Feature Resonance​​ phenomenon: during optimization, unknown in-distribution (ID) samples consistently exhibit larger representation changes than OOD samples even without real label supervision. RSL leverages this by creating a practical micro-level proxy to measure feature vector movement during a single training step and integrating it with synthetic OOD nodes; this combination enables training an effective, label-independent OOD classifier, with theoretical guarantees of superior separability during the resonance period and achieving state-of-the-art performance.

**Questions:**

1. The approach of "aligning the features of known ID nodes to an arbitrary target vector" (in Line 153) appears counterintuitive. ​​Beyond toy datasets, does the method also apply this "align the features of known ID nodes to an arbitrary target vector" strategy using Equation (2) to other datasets?​
2. According to Definition 2 (in line 166-170), identifying t is of crucial importance, as the choice of t is highly sensitive. For different datasets, does the method employ distinct t values?​
3. Section 2.1 and Section 2.2 enhance OOD detection from two distinct perspectives. How are these two components interconnected? Furthermore, does leveraging synthetic OOD nodes to fine-tune the model impact the selection of the critical parameter t defined in Section 2.1?
4. The methodology presented in this paper appears to exhibit limited relevance to graph-specific structures. Have the authors validated their approach on standard image datasets?

If the authors can effectively address my questions, I will consider raising the rating.

**Ethical Concerns:**

["NO or VERY MINOR ethics concerns only"]

**Final Justification:**

The author has addressed most of my concerns. After reviewing the other reviewers' comments and authors' response, I have decided to raise my rating to 4.

**Limitations:**

Please see weakness and questions.

**Quality:**

3

**Strengths And Weaknesses:**

Strengths：
1. ​​Compared to the baseline, the method demonstrates good performance.​​
2. The paper is well-organized, with clear writing logic making it easy to follow.​​
3. ​​The motivation behind the method is good.​

Weaknesses：
1. Although Feature Resonance has novelty to some extent, identifying the appropriate period (in Line 183-184) is critical. While the paper uses the validation set for this, the introduction of the validation set is not specifically elaborated. Furthermore, without a validation set, the method would become inapplicable.
2. The explanation for Equation (4) is insufficient. The paper fails to clearly justify why synthetic OOD nodes need to be generated or why they are synthesized in this specific manner.

---

> ### Author Rebuttal · Authors · 2025-07-30
>
> Thanks very much for your insightful comments and suggestions! Below we address the feedback and comments in detail:
>
> >**W1:** ... the introduction of the validation set is not specifically elaborated. Furthermore, without a validation set, the method would become inapplicable.
> >**A:** **Our use of the validation set strictly follows the setup of our latest baseline EnergyDef [1]**, where the validation and test sets are constructed by randomly splitting the unknown ID and OOD nodes in a 1:2 ratio. In EnergyDef [1], the validation set is used to select the best checkpoint. In our method, the validation set serves **two roles**: during Stage 1, it is used to select the optimal $t$ when mining high-confidence OOD nodes via feature resonance; during Stage 2, it is used to select the best checkpoint for training the OOD classifier. **Importantly, this strategy of selecting the best model based on validation performance is also widely adopted in prior OOD detection works [1-6].**
>
> >**W2:** The explanation for Equation (4) is insufficient.
> >**A:** Thank you for raising this important point. Our decision to generate synthetic OOD nodes via Equation (4) is motivated by two key considerations:
> **First**, although we believe that **feature resonance is a fundamental phenomenon** that offers **valuable insight** into OOD detection, we also recognize the community's strong focus on achieving state-of-the-art performance (a gentle complaint with full respect). Since recent baselines (e.g., EnergyDef [1]) commonly adopt synthetic OOD generation, we include a similar component to ensure fair comparison. Importantly, this integration is not superficial—our method is **orthogonal** to existing techniques and can enhance them by using feature resonance to guide the selection of informative OOD signals, leading to more principled synthesis.
> **Second**,in semi-supervised graph settings, many unlabeled node representations—including real OOD instances—are inherently present. We leverage feature resonance to **identify these nodes with high confidence**, then **synthesize additional OOD nodes guided by them**, as detailed in Equation (4). Compared to heuristic-based baselines like EnergyDef, our method produces samples that **better match real OOD distributions (Appendix Fig. 7)**.
> The effectiveness of this design is supported by Table 2, where training with our synthetic OOD nodes consistently improves performance. Table 7 further shows that **feature resonance reliably selects high-quality real OOD nodes, outperforming other strategies**.
>
> >**Q1:** ...Beyond toy datasets, does the method also apply this "align the features of known ID nodes to an arbitrary target vector" strategy using Equation (2) to other datasets?
> >**A:**
> Yes, **for all datasets**, we consistently align the representations of known ID samples toward an arbitrary fixed vector to induce the feature resonance phenomenon.
>
> >**Q2:** ...For different datasets, does the method employ distinct $t$ values?
> >**A:**
> Thank you for the insightful question! The optimal value of $t$ does vary across datasets; however, in practice, **we do not manually set $t$ for each individual dataset**. Instead, we select the optimal $t$ by evaluating performance on a validation set. **This approach aligns with the common practice in prior OOD detection methods [1-6]**, where validation sets are routinely used to choose the best checkpoint. That said, we also observe that the performance is relatively stable across a reasonable range of $t$ values, especially once it falls within the effective range for most datasets (as shown in Figures 2 and 3). This suggests that $t$ is not overly sensitive in practice, and our method remains robust under typical validation-based selection.
>
> >**Q3-1:** Section 2.1 and Section 2.2 enhance OOD detection from two distinct perspectives. How are these two components interconnected?
> >**A:** Great question! In Section 2.1, we use a toy dataset to intuitively illustrate macro-level feature resonance: unknown ID samples shift more than OOD ones. However, this pattern is not consistent on real datasets (Section 2.2). To address this, we perform a finer-grained, step-wise analysis—termed micro-level feature resonance—**which consistently appears across all real datasets.**
> Although it is challenging to fully explain micro-level feature resonance, we aim to provide some theoretical insights. This aligns with the **Information Bottleneck (IB) theory** [7-8], which suggests that **models initially memorize broad information, then gradually compress irrelevant parts while preserving task-relevant features**.  According to IB, the representation $T$ is optimized by: $$\mathop{\text{min}}\limits_{p(T|X)} I(X;T) -  I(T;Y)$$ where $I(X;T)$ measures how much input information is retained, $I(T;Y)$  indicates task relevance.
> >1. **Early training:** $I(X;T) \uparrow$, $I(T;Y)$ is low $\Rightarrow$ large information redundancy, unstable representations, and little or no resonance;
> >2. **Middle training:** $I(X;T) \downarrow, I(T;Y) \uparrow$ $\Rightarrow$ irrelevant information is compressed, task-relevant features are amplified, resulting in strong feature resonance;
> >3. **Late training:** possible overfitting, $I(X;T) \uparrow$ again, but no further gain in $I(T;Y)$ $\Rightarrow$ representations become more complex, and feature resonance diminishes.
> >
> >This dynamic shows that the compression phase in the middle of training corresponds to a point where irrelevant variation is reduced, allowing feature resonance to become most salient.
>
> >**Q3-2:** Does leveraging synthetic OOD nodes to fine-tune the model impact the selection of the critical parameter $t$ defined in Section 2.1?
> >**A:** **The use of synthetic nodes does not affect the selection of the optimal $t$**, because $t$ is determined in the first stage to select high-confidence real OOD nodes, and these nodes guide the update of synthetic OOD representations in the second stage, the synthetic nodes do not affect the choice of $t$.
>
> >**Q4:** ...Have the authors validated their approach on standard image datasets?
> >**A:**
> Insightful question!  While our method is originally proposed in the context of node-level OOD detection, **the core idea of feature resonance is rooted in representation dynamics**. As per your suggestion, we apply our method to standard image datasets **following the setup in [9] strictly**—using image representations extracted from ResNet-18 models trained on CIFAR-10 with either cross-entropy loss ("without contrastive learning") or supervised contrastive learning ("with contrastive learning"). We induce resonance by aligning known ID features to a random target and measure step-wise changes in unknown samples. As shown in Tables 1 and 2 below, our method remains effective on images. Models with stronger initialization (via contrastive learning) exhibit more pronounced resonance, **consistent with Table 5 of the main paper**. We also evaluate our method on **graph-level OOD detection (Appendix Table 13)**, where representations are independent like in images, and observe similarly strong results—**supporting the universality of the feature resonance phenomenon**.
>
> **Table 1. Results on CIFAR-10.** Comparison with competitive OOD detection methods.  ↑ indicates larger values are better and vice versa.
> |Method|SVHN|SVHN|LSUN|LSUN|iSUN|iSUN|Texture|Texture|Average|Average|
> |-|-|-|-|-|-|-|-|-|-|-|
> || FPR↓|AUROC↑|FPR↓|AUROC↑|FPR↓|AUROC↑|FPR↓|AUROC↑|FPR↓|AUROC↑|
> |||||**Without Contrastive Learning**|
> |MSP|59.66|91.25|45.21|93.80|54.57|92.12|66.45|88.50|56.47|91.42|
> |ODIN|53.78|91.30|10.93|97.93|28.44|95.51|55.59|89.47|37.19|93.55|
> |Energy|54.41|91.22|10.19|98.05|27.52|95.59|55.23|89.37|36.83|93.56|
> |GODIN|18.72|96.10|11.52|97.12|30.02|94.02|33.58|92.20|23.46|94.86|
> |Mahalanobis|9.24|97.80|67.73|73.61|6.02|98.63|23.21|92.91|26.55|90.74|
> |KNN|27.97|95.48|18.50|96.84|24.68|95.52|26.74|94.96|24.47|95.70|
> |FR $\tau$ (ours)|23.50|94.85|11.48|97.80|20.93|95.67|29.22|95.28|21.28|95.90|
> |||||**With Contrastive Learning** |
> |CSI|37.38|94.69|5.88|98.86|10.36|98.01|28.85|94.87|20.62|96.61|94.38|
> |SSD+|1.51|99.68|6.09|98.48|33.60|95.16|12.98|97.70|13.55|97.76|**95.07**|
> |KNN+|2.42|99.52|1.78|99.48|20.06|96.74|8.09|98.56|8.09|98.56|**95.07**|
> |FR $\tau$ (ours)|3.27|99.34|0.44|99.84|9.24|98.23|14.57|97.28|**6.88**|**98.67**|**95.07**|
>
> **Table 2.** Evaluation on hard OOD detection tasks. Model is trained on CIFAR-10 with SupCon loss.
> |Method|LSUN-FIX |LSUN-FIX|LSUN-FIX|ImageNet-FIX|ImageNet-FIX|ImageNet-FIX|ImageNet-R|ImageNet-R|ImageNet-R|Average|Average|Average|
> |-|-|-|-|-|-|-|-|-|-|-|-|-|
> ||AUROC↑|AUPR↑|FPR↓|AUROC↑|AUPR↑|FPR↓|AUROC↑|AUPR↑|FPR↓|AUROC↑|AUPR↑|FPR↓|
> |SSD+|95.52|96.47|29.88|94.85|95.77|32.29|93.40|94.93|45.88|94.59|95.72|36.02|
> |KNN+|96.51|97.20|21.54|95.71|96.37|25.93|95.08|95.95|30.20|95.77|96.51|25.89|
> |FR $\tau$ (ours)|96.41|97.10|21.80|95.13|95.66|26.76|97.33|97.74|15.27|**96.29**|**96.83**|**21.28**|
>
>
> [1] Gong Z, et al. "An energy-centric framework for category-free out-of-distribution node detection in graphs." KDD'24.
> [2] Zheng Y, et al. "Generative and contrastive self-supervised learning for graph anomaly detection." TKDE'21.
> [3] Bergman L, et al. "Classification-based anomaly detection for general data." ICLR'20.
> [4] Qiu C, et al. "Neural transformation learning for deep anomaly detection beyond images." ICML'21
> [5] Zhao X, et al. "Uncertainty aware semi-supervised learning on graph data." NIPS'20
> [6] Song Y, et al. "Learning on graphs with out-of-distribution nodes." KDD'22
> [7] Tishby N, et al. "Deep learning and the information bottleneck principle." ITW'15
> [8] Saxe A M, et al. "On the information bottleneck theory of deep learning." J STAT MECH-THEORY E'19
> [9] Sun Y, et al. "Out-of-distribution detection with deep nearest neighbors." ICML'22

---

> ### Author Response · Authors · 2025-08-07
>
> Dear Reviewer J96B,
>
> We hope this message finds you well.
>
> We sincerely appreciate the time and effort you have dedicated to reviewing our manuscript, as well as your valuable and constructive suggestions. As the discussion period is drawing to a close, we noticed that we have not yet received your feedback on our rebuttal.
>
> Your insights are very important to us, and we would be grateful if you could share any remaining thoughts or questions. Please don’t hesitate to let us know if you require any further clarification regarding our responses.
>
> Thank you once again for your kind contribution.
>
> Best regards,
>
> The Authors

---

> ### Author Response · Authors · 2025-08-08
> **Sincerely look forward to further discussions**
>
> Dear Reviewer, thank you for your valuable comments and suggestions. We have carefully addressed all the reviewers' concerns in our rebuttal, and the other three reviewers, namely **UadC**, **svYR**, and **XXzf**, have responded to the rebuttal and engaged in the discussion with us. We sincerely look forward to receiving your response. Thank you very much!

---

### Official Review · Reviewer_XXzf · 2025-07-02

**Clarity:** 2
**Significance:** 3
**Originality:** 3
**Rating:** 4
**Confidence:** 4

**Summary:**

The authors propose an out-of-distribution (OOD) detection method that does not rely on pretext tasks or label supervision. They identify a new phenomenon, termed Feature Resonance, which emphasizes the feature space instead of the label space. Building on this insight, the authors introduce a novel graph OOD detection framework called RSL. This framework consists of two key components: a micro-level proxy for feature resonance and a synthetic OOD node strategy used to train the OOD classifier. The effectiveness of the proposed approach is demonstrated through experiments on 13 different datasets.

**Questions:**

1.	Why does the micro-level proxy can help measure feature resonance? This is not clearly explained.

**Ethical Concerns:**

["NO or VERY MINOR ethics concerns only"]

**Final Justification:**

The authors have done an effective rebuttal. Thank you.

**Limitations:**

Yes

**Quality:**

3

**Strengths And Weaknesses:**

Strengths:

1.	Investigating a setting that is independent of pretext tasks and label supervision is an interesting and valuable direction.
2.	The intuition behind the feature resonance phenomenon is natural and compelling.
3.	The experiments demonstrate consistent improvements across various settings.

Weaknesses:

1.	Is the Micro-level Feature Resonance Phenomenon commonly observed in real-world scenarios?
2.	Experiments on toy datasets may not accurately reflect real-world conditions. Does the phenomenon shown in Figure 1 also appear in real-world datasets? Please provide the same experiment on real datasets for comparison.
3.	Figure 2 only presents the OOD detection performance, but does not illustrate the feature resonance phenomenon itself.
4.	In Section 2, your method assumes that samples with representations that change rapidly and are similar to known in-distribution (ID) samples are classified as ID, while those that do not change rapidly are considered OOD. Section 2.1 formalizes this phenomenon, and Section 2.2 introduces an arbitrary target to leverage it. However, the rationale behind why this approach works is not sufficiently explained; the paper focuses more on the implementation than on the underlying reasoning.
5.	The technical design in Section 2.2 appears somewhat ad hoc. In Definition 2, why is it necessary to ensure the inequality and define the filtering score in this particular way? Is this the only possible way to define the score? Please clarify the motivation for this specific formulation.
6.	How does Theorem 2 relate to the feature resonance phenomenon? The connection between the theoretical analysis and the design choices is unclear.

---

> ### Author Rebuttal · Authors · 2025-07-30
>
> Thanks very much for your insightful comments and suggestions! Our detailed responses are as follows.
> >**W1:** Is the Micro-level Feature Resonance Phenomenon commonly observed in real-world scenarios?
> >**A:** Feature resonance may be more pronounced in nodes from graphs with high homophily, as node features are reinforced through propagation along edges. However, since feature resonance is defined in the **feature space**, we believe this phenomenon is **generally applicable across different data modalities**. To demonstrate its generality, we apply our method to standard image datasets **following the setup in [1] strictly**——using image representations extracted from ResNet-18 models trained on CIFAR-10 with either cross-entropy loss ("without contrastive learning") or supervised contrastive learning ("with contrastive learning"). We induce resonance by aligning known ID features to a random target and measure step-wise changes in unknown samples. As shown in **Table 1 below**, our method remains effective on images. Models with stronger initialization (via contrastive learning) exhibit more pronounced resonance, **consistent with Table 5 of the main paper**. Further, **Table 2** below shows strong performance on a more challenging image OOD benchmark. We also evaluate our method on **graph-level OOD detection (Appendix Table 13)**, where representations are independent like in images, and observe similarly strong results—**supporting the universality of the feature resonance phenomenon**.
>
> **Table 1. Results on CIFAR-10.** Comparison with competitive OOD detection methods.  ↑ indicates larger values are better and vice versa.
> |Method|SVHN|SVHN|LSUN|LSUN|iSUN|iSUN|Texture|Texture|Average|Average|
> |-|-|-|-|-|-|-|-|-|-|-|
> || FPR↓|AUROC↑|FPR↓|AUROC↑|FPR↓|AUROC↑|FPR↓|AUROC↑|FPR↓|AUROC↑|
> |||||**Without Contrastive Learning**|
> |MSP|59.66|91.25|45.21|93.80|54.57|92.12|66.45|88.50|56.47|91.42|
> |ODIN|53.78|91.30|10.93|97.93|28.44|95.51|55.59|89.47|37.19|93.55|
> |Energy|54.41|91.22|10.19|98.05|27.52|95.59|55.23|89.37|36.83|93.56|
> |GODIN|18.72|96.10|11.52|97.12|30.02|94.02|33.58|92.20|23.46|94.86|
> |Mahalanobis|9.24|97.80|67.73|73.61|6.02|98.63|23.21|92.91|26.55|90.74|
> |KNN|27.97|95.48|18.50|96.84|24.68|95.52|26.74|94.96|24.47|95.70|
> |FR $\tau$ (ours)|23.50|94.85|11.48|97.80|20.93|95.67|29.22|95.28|21.28|95.90|
> |||||**With Contrastive Learning** |
> |CSI|37.38|94.69|5.88|98.86|10.36|98.01|28.85|94.87|20.62|96.61|94.38|
> |SSD+|1.51|99.68|6.09|98.48|33.60|95.16|12.98|97.70|13.55|97.76|**95.07**|
> |KNN+|2.42|99.52|1.78|99.48|20.06|96.74|8.09|98.56|8.09|98.56|**95.07**|
> |FR $\tau$ (ours)|3.27|99.34|0.44|99.84|9.24|98.23|14.57|97.28|**6.88**|**98.67**|**95.07**|
>
> **Table 2.** Evaluation on hard OOD detection tasks. The model is trained on CIFAR-10 with SupCon loss.
> |Method|LSUN-FIX |LSUN-FIX|LSUN-FIX|ImageNet-FIX|ImageNet-FIX|ImageNet-FIX|ImageNet-R|ImageNet-R|ImageNet-R|Average|Average|Average|
> |-|-|-|-|-|-|-|-|-|-|-|-|-|
> ||AUROC↑|AUPR↑|FPR↓|AUROC↑|AUPR↑|FPR↓|AUROC↑|AUPR↑|FPR↓|AUROC↑|AUPR↑|FPR↓|
> |SSD+|95.52|96.47|29.88|94.85|95.77|32.29|93.40|94.93|45.88|94.59|95.72|36.02|
> |KNN+|96.51|97.20|21.54|95.71|96.37|25.93|95.08|95.95|30.20|95.77|96.51|25.89|
> |FR $\tau$ (ours)|96.41|97.10|21.80|95.13|95.66|26.76|97.33|97.74|15.27|**96.29**|**96.83**|**21.28**|
>
> >**W2 & W3:** Does the phenomenon shown in Figure 1 also appear in real-world datasets? Figure 2 does not illustrate the feature resonance phenomenon itself.
> >**A:**  While toy datasets offer the **clearest and most intuitive illustration** of the feature resonance phenomenon, we have conducted extensive experiments on real-world benchmarks (Amazon, Squirrel, WikiCS, YelpChi, Reddit). These show **consistent micro-level resonance patterns**, where ID and OOD nodes differ in representation dynamics across training.
> Although macro-level resonance (Figure 1) may be less visible in real data due to noise, micro-level resonance is stable and general across all datasets (please refer to Figure 4 of the Appendix). Figure 2 and Figure 3 (Appendix) visualize these effects as real-data counterparts to the toy example.
> Regarding Figure 2, the performance metrics are based on the relative representation shift of unknown ID vs. OOD samples. **Therefore, Figure 2 implicitly reflects the micro-level feature resonance phenomenon.** Since **this year's NeurIPS rebuttal does not allow PDF uploads**, we regret that we are unable to present the additional visualizations. We sincerely apologize for this limitation and **will make sure to include a Figure 1-style visualization on real datasets in the revised version.**
>
> >**W4:** ... However, the rationale behind why this approach works is not sufficiently explained.
> >**A:** Thank you for your insightful comment. The rationale behind our approach is grounded in the observation that **ID samples tend to co-evolve in the feature space due to shared semantic structure**, whereas OOD samples lack such alignment and exhibit weaker representation dynamics, leading to measurable differences in representation shift.
> Although it is challenging to fully explain why feature resonance is most prominent in the middle stages of training, we aim to provide some theoretical insights. This aligns with the **Information Bottleneck (IB) theory** [2-3], which suggest that **models initially memorize broad information, then gradually compress irrelevant parts while preserving task-relevant features**. This compression phase in the middle of training corresponds to a point where irrelevant variation is reduced, allowing feature resonance to become most salient. According to IB, the representation $T$ is optimized by: $$\mathop{\text{min}}\limits_{p(T|X)} I(X;T) -  I(T;Y)$$ where $I(X;T)$ measures how much input information is retained, $I(T;Y)$ indicates task relevance.
> >1. **Early training:** $I(X;T) \uparrow$, $I(T;Y)$ is low $\Rightarrow$ large information redundancy, unstable representations, and little or no resonance;
> >2. **Middle training:** $I(X;T) \downarrow, I(T;Y) \uparrow$ $\Rightarrow$ irrelevant information is compressed, task-relevant features are amplified, resulting in strong feature resonance;
> >3. **Late training:** possible overfitting, $I(X;T) \uparrow$ again, but no further gain in $I(T;Y)$ $\Rightarrow$ representations become more complex, and feature resonance diminishes.
> >
> >This dynamic explains why feature resonance tends to emerge most clearly during the middle stages of training. We will clarify this motivation and the theoretical grounding more explicitly in the revised version.
>
> >**W5:**  In Definition 2, why is it necessary to ensure the inequality and define the filtering score in this particular way? Is this the only possible way to define the score?
> >**A:** Thank you for raising these questions. The filtering score in Definition 2 is designed to quantify the relative feature variation of each unknown node during training by leveraging the feature resonance phenomenon. The intuition is as follows:
>  According to our observations, unknown ID samples tend to exhibit stronger and more consistent feature updates during training compared to OOD samples.
>  Therefore, by measuring how much a sample's representation changes across training epochs, we can estimate its likelihood of being an ID sample.
>  While this formulation may not be the only possible one, **it is simple, effective, and empirically validated across multiple datasets** (see **Table 7** of the main paper for comparisons with alternative designs).
>  We also chose this formulation for its interpretability—**it directly reflects the dynamic behavior of representations over time, which is central to the resonance effect.**
>
> >**W6:**  How does Theorem 2 relate to the feature resonance phenomenon?
> >**A:**  Great question! Theorem 2 provides a theoretical justification for the timing of the feature resonance phenomenon. Specifically, it shows that **the upper bound of the filtering error related to micro-level feature changes decreases and then increases over training steps**, implying that there exists a middle stage during training where the feature representations of ID and OOD samples diverge most clearly. **This directly supports our design choice of leveraging micro-level feature resonance in a step-wise manner**, especially during this critical middle phase, for more reliable OOD detection.
>
> >**Q1:**  Why does the micro-level proxy can help measure feature resonance?
> >**A:** Thank you for the thoughtful question. The micro-level proxy captures fine-grained, step-wise changes in representations during training. Intuitively, **feature resonance means some samples—especially unknown ID ones—react more strongly to optimization aligned with known ID patterns.** Our proxy measures this by tracking the magnitude of feature shifts over training steps. Empirically, unknown ID samples show larger representation changes than OOD samples under the same inductive bias. Thus, the micro-level proxy effectively indicates feature resonance, even without multi-class label information.
>
> [1] Sun Y, et al. "Out-of-distribution detection with deep nearest neighbors."ICML'22
> [2] Tishby N, et al. "Deep learning and the information bottleneck principle." ITW'15
> [3] Saxe A M, et al. "On the information bottleneck theory of deep learning." J STAT MECH-THEORY E'19

---

> > ### Comment · Reviewer_XXzf · 2025-08-06
> >
> > Thank you for your effort in resolving my major concerns. I will keep my positive score.

---

> > > ### Author Response · Authors · 2025-08-06
> > >
> > > Thanks for your valuable suggestions that make our paper more solid. We will incorporate the new results and the fruitful points in our new revision.

---

### Official Review · Reviewer_UadC · 2025-07-02

**Clarity:** 4
**Significance:** 3
**Originality:** 4
**Rating:** 5
**Confidence:** 5

**Summary:**

This paper tries to address the problem of out-of-distribution node detection and proposes a feature-space-based approach. The method leverages the fact that graphs, which are commonly encountered in semi-supervised scenarios, exhibit a phenomenon during node feature training: at certain training steps, the representations of nodes belonging to the same class as those in the training set undergo larger representation shifts. The authors refer to this as the feature resonance phenomenon and use it as the basis for identifying OOD nodes. In addition, the paper incorporates weak supervision and synthetic OOD node generation to train an OOD node classifier, further enhancing the detection performance.

**Questions:**

See the weaknesses mentioned above.

**Ethical Concerns:**

["NO or VERY MINOR ethics concerns only"]

**Final Justification:**

It has addressed my concerns, and I will keep my score.

**Limitations:**

No, but the authors have proposed a potential solution to be explored in future work.

**Paper Formatting Concerns:**

No.

**Quality:**

4

**Strengths And Weaknesses:**

Strengths:

1. This paper introduces a novel feature resonance phenomenon from the feature space perspective. What’s especially interesting is that this phenomenon seems to happen no matter what training setup you use — it’s not tied to specific paradigms or whether labels are available. This makes it potentially useful in both supervised and unsupervised scenarios.

2. The paper provides a thorough theoretical proof for the upper bound of the proposed method's error. I went through the proof and didn’t spot any major issues.

3. The experiments are comprehensive, covering different setups.

4. It is well-written and easy to understand.

Weaknesses :

1. The paper does not seem to explain why the feature resonance phenomenon tends to occur in the middle stages of training.

2. Is the feature resonance phenomenon exclusive to nodes? Does it also occur in other types of data?

3. Compared to the synthetic OOD node generation method in [1], what are the specific advantages of the synthetic OOD node generation method proposed in this paper?

4. What are the practical application scenarios for this method?

[1] Gong, Zheng, and Ying Sun. "An energy-centric framework for category-free out-of-distribution node detection in graphs." Proceedings of the 30th ACM SIGKDD Conference on Knowledge Discovery and Data Mining. 2024.

---

> ### Author Rebuttal · Authors · 2025-07-30
>
> Thank you very much for your comments and suggestions! We are happy you enjoyed the paper. Below are our detailed responses:
> > **W1:**
> The paper does not seem to explain why the feature resonance phenomenon tends to occur in the middle stages of training.
> >**A:**
> We appreciate the reviewer's question. Empirically, we observe that feature resonance peaks in the middle of training. Although it is challenging to fully explain why feature resonance is most prominent in the middle stages of training, we aim to provide some theoretical insights. This aligns with the **Information Bottleneck (IB) theory** [1-2] and recent feature learning studies [3-4], which suggest that **models initially memorize broad information, then gradually compress irrelevant parts while preserving task-relevant features**—reflecting an emerging inductive bias. This compression phase in the middle of training corresponds to a point where irrelevant variation is reduced, allowing feature resonance to become most salient. According to IB, the representation $T$ is optimized by:
> $$\mathop{\text{min}}\limits_{p(T|X)} I(X;T) -  I(T;Y)$$
> where $I(X;T)$ measures how much input information is retained, $I(T;Y)$  indicates task relevance.
> >1. **Early training:** $I(X;T) \uparrow$, $I(T;Y)$ is low $\Rightarrow$ large information redundancy, unstable representations, and little or no resonance;
> >2. **Middle training:** $I(X;T) \downarrow, I(T;Y) \uparrow$ $\Rightarrow$ irrelevant information is compressed, task-relevant features are amplified, resulting in strong feature resonance;
> >3. **Late training:** possible overfitting, $I(X;T) \uparrow$ again, but no further gain in $I(T;Y)$ $\Rightarrow$ representations become more complex, and feature resonance diminishes.
> >
> >This dynamic explains why feature resonance tends to emerge most clearly during the middle stages of training.
>
> > **W2:**
> Is the feature resonance phenomenon exclusive to nodes? Does it also occur in other types of data?
> >**A:** Feature resonance may be more pronounced in nodes from graphs with high homophily, as node features are reinforced through propagation along edges. However, since feature resonance is defined in the feature space, we believe **it naturally generalizes across different data modalities**. To validate this, we apply our method to standard image datasets **following the setup in [5] strictly**——using image representations extracted from ResNet-18 models trained on CIFAR-10 with either cross-entropy loss ("without contrastive learning") or supervised contrastive learning ("with contrastive learning"). We induce resonance by aligning known ID features to a random target and measure step-wise changes in unknown samples. As shown in **Table 1 below**, our method remains effective on images. Models with stronger initialization (via contrastive learning) exhibit more pronounced resonance, **consistent with Table 5 of the main paper**. Further, **Table 2 below** shows strong performance on a more challenging image OOD benchmark. We also evaluate our method on **graph-level OOD detection (Appendix Table 13)**, where representations are independent like in images, and observe similarly strong results—**supporting the universality of the feature resonance phenomenon**.
>
>
> **Table 1. Results on CIFAR-10.** Comparison with competitive OOD detection methods.  ↑ indicates larger values are better and vice versa.
> |Method|SVHN|SVHN|LSUN|LSUN|iSUN|iSUN|Texture|Texture|Average|Average|
> |-|-|-|-|-|-|-|-|-|-|-|
> || FPR↓|AUROC↑|FPR↓|AUROC↑|FPR↓|AUROC↑|FPR↓|AUROC↑|FPR↓|AUROC↑|
> |||||**Without Contrastive Learning**|
> |MSP|59.66|91.25|45.21|93.80|54.57|92.12|66.45|88.50|56.47|91.42|
> |ODIN|53.78|91.30|10.93|97.93|28.44|95.51|55.59|89.47|37.19|93.55|
> |Energy|54.41|91.22|10.19|98.05|27.52|95.59|55.23|89.37|36.83|93.56|
> |GODIN|18.72|96.10|11.52|97.12|30.02|94.02|33.58|92.20|23.46|94.86|
> |Mahalanobis|9.24|97.80|67.73|73.61|6.02|98.63|23.21|92.91|26.55|90.74|
> |KNN|27.97|95.48|18.50|96.84|24.68|95.52|26.74|94.96|24.47|95.70|
> |FR $\tau$ (ours)|23.50|94.85|11.48|97.80|20.93|95.67|29.22|95.28|21.28|95.90|
> |||||**With Contrastive Learning** |
> |CSI|37.38|94.69|5.88|98.86|10.36|98.01|28.85|94.87|20.62|96.61|94.38|
> |SSD+|1.51|99.68|6.09|98.48|33.60|95.16|12.98|97.70|13.55|97.76|**95.07**|
> |KNN+|2.42|99.52|1.78|99.48|20.06|96.74|8.09|98.56|8.09|98.56|**95.07**|
> |FR $\tau$ (ours)|3.27|99.34|0.44|99.84|9.24|98.23|14.57|97.28|**6.88**|**98.67**|**95.07**|
>
> **Table 2.** Evaluation on hard OOD detection tasks. Model is trained on CIFAR-10 with SupCon loss.
> |Method|LSUN-FIX |LSUN-FIX|LSUN-FIX|ImageNet-FIX|ImageNet-FIX|ImageNet-FIX|ImageNet-R|ImageNet-R|ImageNet-R|Average|Average|Average|
> |-|-|-|-|-|-|-|-|-|-|-|-|-|
> ||AUROC↑|AUPR↑|FPR↓|AUROC↑|AUPR↑|FPR↓|AUROC↑|AUPR↑|FPR↓|AUROC↑|AUPR↑|FPR↓|
> |SSD+|95.52|96.47|29.88|94.85|95.77|32.29|93.40|94.93|45.88|94.59|95.72|36.02|
> |KNN+|96.51|97.20|21.54|95.71|96.37|25.93|95.08|95.95|30.20|95.77|96.51|25.89|
> |FR $\tau$ (ours)|96.41|97.10|21.80|95.13|95.66|26.76|97.33|97.74|15.27|**96.29**|**96.83**|**21.28**|
>
>
> > **W3:** Compared to the synthetic OOD node generation method in EnergyDef, what are the specific advantages of the synthetic OOD node generation method proposed in this paper?
> >**A:** Great question! Although EnergyDef [6] also uses Langevin dynamics to synthesize OOD nodes, the OOD nodes generated in this way are likely to **deviate significantly from the true OOD distribution**. Our method, on the other hand, fully leverages the semi-supervised nature of graphs—we have access to nodes with known features but unknown labels. By using feature resonance, we first **identify a set of high-confidence real OOD nodes as references**, resulting in synthesized OOD nodes that **better match the true OOD distribution**. The feature visualization in **Figure 7 of the appendix** confirms this, showing that the node representations trained with our method achieve better separation between ID and OOD nodes.
>
> > **W4:**  What are the practical application scenarios for this method?
> >**A:**
> Since feature resonance does not depend on specific proxy tasks or multi-class label supervision, we believe it can be readily generalized to a wide range of common data filtering applications, including anomaly detection and out-of-distribution detection across arbitrary numbers of categories.
>
> [1] Tishby N, Zaslavsky N. Deep learning and the information bottleneck principle[C]//2015 ieee information theory workshop (itw). Ieee, 2015: 1-5.
> [2] Saxe A M, Bansal Y, Dapello J, et al. On the information bottleneck theory of deep learning[J]. Journal of Statistical Mechanics: Theory and Experiment, 2019, 2019(12): 124020.
> [3] Allen-Zhu Z, Li Y. Feature purification: How adversarial training performs robust deep learning[C]//2021 IEEE 62nd annual symposium on foundations of computer science (FOCS). IEEE, 2022: 977-988.
> [4] Cao Y, Chen Z, Belkin M, et al. Benign overfitting in two-layer convolutional neural networks[J]. Advances in neural information processing systems, 2022, 35: 25237-25250.
> [5] Sun Y, Ming Y, Zhu X, et al. Out-of-distribution detection with deep nearest neighbors[C]//International conference on machine learning. PMLR, 2022: 20827-20840.
> [6] Gong, Zheng, and Ying Sun. "An energy-centric framework for category-free out-of-distribution node detection in graphs." Proceedings of the 30th ACM SIGKDD Conference on Knowledge Discovery and Data Mining. 2024.

---

> > ### Comment · Area_Chair_kkwu · 2025-08-05
> >
> > Dear Reviewer UadC,
> >
> > Thank you for your effort in the review.
> >
> > As the discussion period ends soon, could you check authors rebuttals and see if they addressed your concern?
> >
> > If authors have resolved your questions, do tell them so.
> >
> > If authors have not resolved your questions, do tell them so too.
> >
> > Thanks.
> >
> > AC

---

> > ### Comment · Reviewer_UadC · 2025-08-08
> >
> > Thank you for the response. It has addressed my concerns, and I will keep my score.

---

> > > ### Author Response · Authors · 2025-08-08
> > >
> > > Thanks for your reply! We also sincerely thank you for your valuable time on our paper and thank you for supporting our paper.

---

> ### Author Response · Authors · 2025-08-07
>
> Dear Reviewer UadC,
>
> We hope this message finds you well.
>
> We sincerely appreciate the time and effort you have dedicated to reviewing our manuscript, as well as your valuable and constructive suggestions. As the discussion period is drawing to a close, we noticed that we have not yet received your feedback on our rebuttal.
>
> Your insights are very important to us, and we would be grateful if you could share any remaining thoughts or questions. Please don’t hesitate to let us know if you require any further clarification regarding our responses.
>
> Thank you once again for your kind contribution.
>
> Best regards,
>
> The Authors

---

### Author Response · Authors · 2025-08-09
**A summary of the key pros and concerns for the convenience of (senior) area chairs and reviewers**

Dear (Senior) ACs and Reviewers,

We extend our sincere gratitude for the time and effort you have dedicated to reviewing our manuscript. We highly value the thoughtful and constructive feedback from all reviewers, which has greatly contributed to improving the clarity, rigor, and overall quality of our work.

**Feature resonance in deep learning and its role in OOD detection are analyzed for the first time, accompanied by a new method leveraging this phenomenon.** We hope our work will inspire further research in this important area.

We appreciate the reviewers’ recognition of the novelty, theoretical grounding, and empirical rigor of our work. For your convenience, we summarize the key pros and concerns raised by the reviewers, alongside our responses:

>**Key pros noted by the reviewers:**
>**P1:** The paper identifies a **novel and interesting feature resonance phenomenon** with a **natural and compelling intuition**, applicable across various training setups independent of pretext tasks, making it an **interesting and valuable direction** (_All Reviewers_ **UadC**, **XXzf**, **J96B**, **svYR**).
>**P2:** The theoretical analysis is rigorous and well-founded (Reviewers **UadC**, **svYR**).
>**P3:** The experiments are comprehensive and demonstrate consistent improvements across diverse settings (Reviewers **UadC**, **XXzf**, **J96B**).
>**P4:** The paper is well-written, with clear motivation and effective visualizations (Reviewers **UadC**, **J96B**, **svYR**).

>**Key concerns noted by the reviewers and our responses:**
>**C1:** _Clarification on the universality of feature resonance in real-world and non-graph data settings._
We have clarified through additional experiments provided during the rebuttal period that **feature resonance is a general phenomenon** observed not only in graph data but also in other domains such as images, highlighting the significance of this finding (Reviewers **UadC**, **XXzf**, **J96B**, **svYR**).
>**C2:** _Why is micro-level feature resonance more pronounced?_
Starting from the **information bottleneck theory**, we explain that micro-level feature resonance is most significant during the information compression phase in the middle of training (Reviewers **UadC**, **XXzf**, **J96B**).
>**C3:** _Validation protocols and hyperparameter selection._
We confirm and clarify that all baselines and our method were tuned using **the same validation sets with standardized protocols [1]** to ensure the fair comparison (Reviewers **J96B**, **svYR**).
>[1] Gong Z, et al. "An energy-centric framework for category-free out-of-distribution node detection in graphs." KDD'24.

We would also like to note that we have addressed the concerns raised by three reviewers—**UadC**, **XXzf**, and **svYR**—to their satisfaction. **Unfortunately, to date, we have not received a follow-up from Reviewer J96B.** **_However, most of the concerns raised by Reviewer J96B substantially overlap with those from other reviewers, and our detailed responses and clarifications have been well received by them._**

Thank you very much for your time and consideration. We hope this will be helpful!

Best regards,

Authors

---

### Decision · Program_Chairs · 2025-09-17

**Decision:**

Accept (poster)

**Comment:**

The paper addresses OOD node detection in graphs. One major challenge is to devise a scalable OOD detection method independent of both pretext tasks and label supervision. The authors formulate a new concept called Feature Resonance, which focuses on the feature space rather than the label space. Subsequently, the authors propose a Resonance-based Separation and Learning (RSL) method. Both theoretical and empirical results demonstrate the validity of the proposed approach.

Strengths:
- The feature resonance notion is novel and a promising direction to explore.
- Solid, well-founded theoretical analysis
- Comprehensive experimental results.

Weaknesses:
- The feature resonance phenomenon on toy datasets may not reflect real-world scenarios, raising questions about its generality.
- The validation process is not clearly explained.
- Experimental setup needs to be more detailed

Overall, there is general consensus among reviewers on the strengths of the paper. The weaknesses can be clarified or explained in the camera ready, as they have already been addressed during rebuttal as acknowledged by all reviewers.